# Population genomics and antimicrobial resistance dynamics of *Escherichia coli* in wastewater and river environments

Jose F. Delgado-Blas [1], Cristina M. Ovejero[1], Sophia David[2], Natalia Montero[1], William Calero-Caceres [3,5], M. Pilar Garcillan-Barcia [4], Fernando de la Cruz [4], Maite Muniesa [3], David M. Aanensen[2] & Bruno Gonzalez-Zorn [1✉]

Aquatic environments are key niches for the emergence, evolution and dissemination of antimicrobial resistance. However, the population diversity and the genetic elements that drive the dynamics of resistant bacteria in different aquatic environments are still largely unknown. The aim of this study was to understand the population genomics and evolutionary events of *Escherichia coli* resistant to clinically important antibiotics including aminoglycosides, in anthropogenic and natural water ecosystems. Here we show that less different *E. coli* sequence types (STs) are identified in wastewater than in rivers, albeit more resistant to antibiotics, and with significantly more plasmids/cell (6.36 vs 3.72). However, the genomic diversity within *E. coli* STs in both aquatic environments is similar. Wastewater environments favor the selection of conserved chromosomal structures associated with diverse flexible plasmids, unraveling promiscuous interplasmidic resistance genes flux. On the contrary, the key driver for river *E. coli* adaptation is a mutable chromosome along with few plasmid types shared between diverse STs harboring a limited resistance gene content.

[1] Antimicrobial Resistance Unit (ARU), Animal Health Department, Faculty of Veterinary Medicine and VISAVET Health Surveillance Centre, Complutense University of Madrid, Madrid, Spain. [2] Centre for Genomic Pathogen Surveillance (CGPS), Wellcome Sanger Institute, Hinxton, UK. [3] Department of Genetics, Microbiology and Statistics, Faculty of Biology, University of Barcelona, Barcelona, Spain. [4] Institute of Biomedicine and Biotechnology (IBBTEC), CSIC, University of Cantabria, Santander, Spain. [5]Present address: UTA RAM One Health, Faculty of Food Science, Engineering and Biotechnology, Technical University of Ambato, Ambato, Ecuador. ✉email: bgzorn@ucm.es

Aquatic environments are one of the most important ecological niches on Earth, as numerous biological phenomena take place in the multiple phases of the water cycle[1]. Water phases and stages constitute genetic reactors, ecological scenarios where environmental conditions lead bacterial evolution due to biological connection, variation, and selection[2]. This situation is especially notorious in wastewater treatment plants (WWTPs), since they collect residual waters from diverse origins and populations where distinct anthropogenic activities occur, including nosocomial environments. Thus, great bacterial diversity and its gene content have the optimal conditions to interact and generate novel associations. The WWTP bacterial populations can eventually reach natural environments and their native bacterial populations, which act as important reservoirs and vehicles for antibiotic resistance mechanisms[2].

Freshwater is one of the natural habitats with the richest diversity of bacterial populations, not only taking into account the species level but also the lineages and clusters of the same species[1]. These indigenous populations also present their own genetic resistance content, although data about antibiotic resistance in natural niches is still scarce[3]. Selected antibiotic resistance mechanisms from previous stages of the water cycle or resistance genes from typically natural bacteria can disseminate through this cycle and reach bacterial species able to colonize and infect human and animal populations, being a high risk for public health[4].

The emergence of carbapenem-resistant *Enterobacteriaceae* (CRE), and their association with other mechanisms conferring resistance to last-line antibiotics such as colistin, highlighted the relevance of aminoglycosides[5]. Aminoglycosides are antimicrobial compounds with bactericidal activity. They interact with the A-site of the 16S rRNA of the 30S ribosomal subunit, leading to an alteration of the bacterial translational process. This class of antibiotics was included in the critically important antimicrobials category of human medicines[6]. Aminoglycosides belonging to the 4,6-disubstituted 2-deoxystreptamine (4,6-DOS) group, which have particular clinical relevance, are widely used for the treatment of complicated infections[5].

Resistance to aminoglycosides can be mediated by diverse mechanisms. Of these, the most worrisome is the posttranscriptional modification of the 16S rRNA by acquired methyltransferase enzymes, which methylate particular residues of the bacterial ribosome, blocking the attachment of aminoglycosides to their cellular target. The 16S rRNA methyltransferases (16S-RMTases) include several enzymes (ArmA, RmtA-RmtH, and NpmA), which are typically found integrated into mobile genetic elements and associated with various plasmid types. ArmA and RmtB are the most prevalent ones worldwide and they completely obliterate the activity of 4,6-DOS aminoglycosides, conferring high resistance levels to the clinically most relevant aminoglycosides[5]. These enzymes are even able to abolish the effect of plazomicin, a novel aminoglycoside considered a last resource compound for the treatment of infections caused by multidrug-resistant bacteria[7]. A few studies have pointed at aquatic environments as important reservoirs for bacteria and mobile genetic elements involved in 16S-RMTase production and dissemination, specially wastewater. However, the ecological scenario of 16S-RMTases in different water environments, their epidemiological associations, and their potential implications for public health remain largely ununderstood[3,8].

The aim of this work was to unravel the genomic structure and population diversity of *Escherichia coli* harboring 16S-RMTase genes from wastewater (WWTPs) and freshwater (rivers), in order to understand the bacterial elements and phenomena that led to the emergence, evolution, and dissemination of the gene-plasmid-bacteria associations in different aquatic environments.

## Results

**Escherichia coli is amongst the dominant pan-aminoglycoside-resistant bacteria from wastewater and river water.** A total of 168 bacteria highly resistant to aminoglycosides were obtained by growing collected samples in MacConkey media supplemented with high aminoglycoside concentrations, from both WWTPs (66 isolates) and rivers (102 isolates) sampled at the same period of time and located in the Barcelona region. Most of the isolates were identified as *Escherichia coli*, being the predominant *Enterobacteriaceae* species in WWTPs (50 isolates, 75.76%) and the second most prevalent in rivers (20 isolates, 19.61%). *Klebsiella pneumoniae* was the predominant bacterium identified in river environments (50 isolates, 49.02%), although its presence in WWTP samples was scarce (7 isolates, 10.61%). The remaining aminoglycoside-resistant species, members of genus *Enterobacter*, *Citrobacter* and *Aeromonas* among others, represented a low fraction in WWTPs (9 isolates, 13.64%) comparing to river environments (32 isolates, 31.37%), showing a higher bacterial diversity of natural over WWTP niches (Fig. 1, Supplementary Data 1).

**Antimicrobial resistance in *E. coli* varies between wastewater and river water environments.** Resistance to diverse antibiotic compounds was related to the sampling origin and the species of the isolates. All isolates were highly resistant to 4,6-DOS aminoglycosides, since they were selected according to their resistance to this family of antibiotics. The 16S-RMTase gene identified in most wastewater isolates was *rmtB* (55 isolates, 83.33%), whereas *armA* was the responsible in the remainder (11 isolates, 16.67%). On the contrary, 100% of isolates from river environments harbored the *armA* gene (Fig. 1, Supplementary Data 1). No other 16S-RMTase gene was detected neither in wastewater nor river isolates. Bacteria from wastewater samples showed higher MIC values to all tested antibiotics (e.g., 77.3% isolates resistant to cefotaxime) than bacteria from freshwater environments (e.g., 0% isolates resistant to cefotaxime), even when comparing isolates belonging to the same species. This was also the case for *E. coli*, where isolates from WWTPs were among the most resistant (e.g., 90% isolates resistant to cefotaxime) while those from rivers were among the most susceptible ones (e.g., 0% isolates resistant to cefotaxime), highlighting key differences between bacteria from these different ecological niches (Supplementary Data 1).

**Distinct *E. coli* populations exist in wastewater and river water environments.** Pulsed-field gel electrophoresis (PFGE) of *E. coli* isolates revealed the existence of two highly prevalent pulsotypes present in the two WWTPs (31 isolates, 62%), and a total of 5 characterized pulsotypes (≥90% similarity in the PFGE pattern) from wastewater samples. Interestingly, 15 isolates (30%) from both WWTPs could not be typed by PFGE. In contrast, river environments showed higher clonal diversity, with up to 10 different pulsotypes characterized, despite the lower number of *E. coli* isolates recovered from these samples (Supplementary Fig. 1). Of the 70 *E. coli* isolates, 43 were selected for further analyses and sequencing by Illumina technology (25 from WWTPs and 18 from river environments), ensuring wide representation in relation to the origins, clonal relatedness, and antibiogram resistance profiles.

Multi-locus sequence typing (MLST) results were highly correlated with PFGE profiles. Almost all *E. coli* isolates from wastewater samples (23 isolates, 92%) belonged to two predominant sequence types (STs) (Fig. 2). The most prevalent one, ST1196/ST632 (Warwick/Pasteur MLST scheme), encompassed the two main pulsotypes identified by PFGE. ST1196 is an

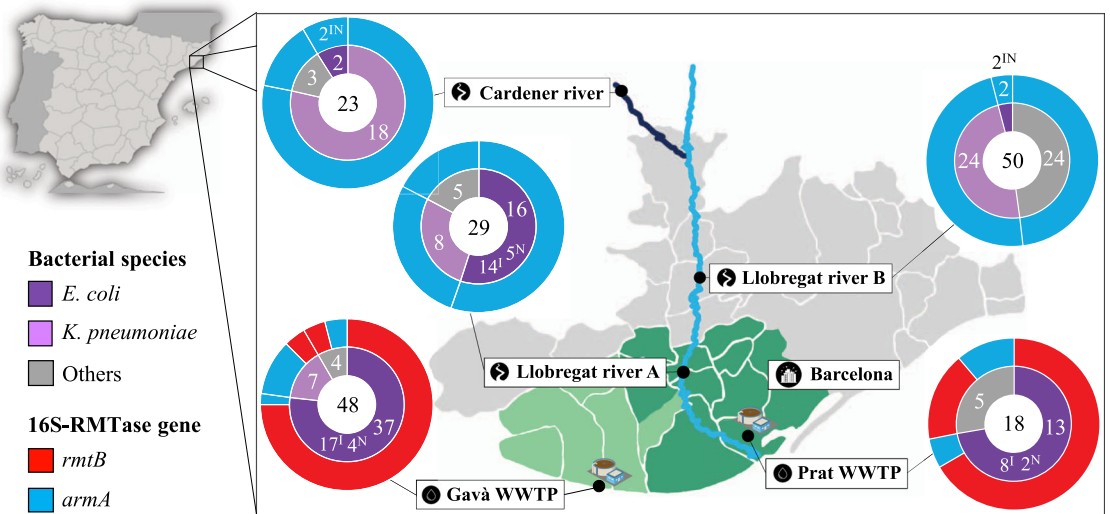

**Fig. 1 Geographical distribution of sampled points in the region of Barcelona (Spain).** The location of Barcelona city is indicated by black circled city icon. The Llobregat river is presented in light blue and the Cardener river in dark blue. The three river sampled locations are indicated by black circled river icon. The two sampled WWTPs are indicated by black circled droplet icon. The collection area of the El Prat WWTP is highlighted in dark green, whereas the area of the Gavà WWTP is highlighted in pale green, collecting the wastewater of 2,000,000 and 370,000 inhabitants, respectively. Total number of pan-aminoglycoside-resistant bacteria collected from each sampling location is indicated in the center of the sunburn diagrams. Inner rings represent the proportion of different pan-aminoglycoside-resistant bacterial species identified in each sampling location, indicating the total number of *E. coli*, *K. pneumoniae* and other species, and the number of *E. coli* isolates selected for later Illumina ([I]) and Nanopore sequencing ([N]). Outer rings show the 16S-RMTase gene harbored by these bacteria.

increasingly prevalent ST related to OXA-48-carbapenemase production and associated with the *mcr-1* colistin-resistance gene expansion in clinical settings[9], hospital wastewater[10] and companion animals[11]. Minimal inhibitory concentration (MIC) values of the isolates belonging to this ST showed resistance to numerous antibiotic classes, including clinically critical antibiotics such as 3[rd] generation cephalosporins and colistin[6]. The other predominant ST in WWTPs, ST224/ST479, comprised all the isolates which were non-typeable by PFGE. ST224 is a pandemic multi-drug resistant ST previously associated with NDM-, CTX-M- and KPC-carbapenemase production, found in both clinical human[12] and animal[13] samples, but also in natural environments[14]. ST224 *E. coli* isolates identified exhibited a common resistance profile, including resistance to 3[rd] generation cephalosporins. However, two isolates from river samples belonging to this same ST showed a different resistance pattern, revealing the influence of the niche in the bacterial resistance phenotype, even within the same ST. One of the *E. coli* isolates from El Prat WWTP was also identified as ST131, a clonal group present in multiple environments with a plethora of resistance mechanisms and virulence factors, representing a major public health concern[15]. River-related *E. coli* showed a higher number of *E. coli* STs compared to wastewater isolates, comprising up to 6 different *E. coli* STs (Fig. 2). All river *E. coli* STs presented similar MIC values for most of the tested antibiotics, which were lower than those exhibited by wastewater *E. coli*. ST607 (Warwick MLST scheme) was the most prevalent ST found in rivers. This ST has been scarcely reported, although it has been already detected in river sediments[16]. MICs of plazomicin for all isolates, both from wastewater and river environments, were ≥512 mg/L, demonstrating the high-level resistance to aminoglycosides conferred by 16S-RMTases, even to an aminoglycoside that has not yet been approved for clinical use in the EU (Fig. 2).

Resistome analysis results were highly correlated with resistance phenotypic profiles. Thus, isolates from WWTPs harbored an heterogenous antibiotic resistance gene content depending on the *E. coli* ST, such as the strong association between the *mcr-1*

gene and isolates belonging to ST1196[17]. This heterogenicity was even present between isolates belonging to ST1196, which exhibited different resistance levels to chloramphenicol and tetracycline depending on the presence of *cmlA1/floR* and *tet (A)*, respectively. However, all wastewater isolates showed a common high-level resistance to β-lactam compounds, including third generation cephalosporins, which could be attributed to the presence of $bla_{CTX-M-55}$ (ST1196) and $bla_{CMY-2}$ (ST224) (Fig. 2). On the contrary, all isolates from river environments possessed a uniform antibiotic resistance gene content, despite comprising different STs (Fig. 2). Almost all of them (17 isolates, 94.4%) were susceptible to β-lactams, including the isolates belonging to ST224, contrasting with wastewater isolates, which have been under anthropogenic pressure, such as clinical treatments with a combination of aminoglycosides and β-lactams[18], that led to this resistance associations. Likewise, plasmidome analysis revealed that the total plasmid content, based on the plasmid incompatibility groups, was heterogeneously distributed among isolates from WWTPs according to *E. coli* STs and closely correlated with the resistance gene content (an average of 6.36 different plasmid replicons per isolate). ST1196 isolates carried a higher plasmid content, exhibiting different plasmid profiles among them. The only plasmid incompatibility group carried by all wastewater isolates was IncFII type, specifically a pC15-1a-like plasmid (Fig. 2). Considering river *E. coli* STs, the total plasmid content was generally lower (an average of 3.72 different plasmid replicons per isolate) and more uniform compared to wastewater STs, similar to the pattern of the resistome (these differences are addressed in the *E. coli* pan-genome structure and genome, plasmid and antibiotic resistance gene diversities section). The most prevalent plasmid incompatibility group found among them was IncHI2A, present in all river isolates except ST224, which harbored a completely different plasmid content (Fig. 2).

**E. coli STs have a similar genomic complexity in wastewater and river water, but the diversity of plasmids and resistance genes is higher in wastewater STs.** The pan-genome of all 43

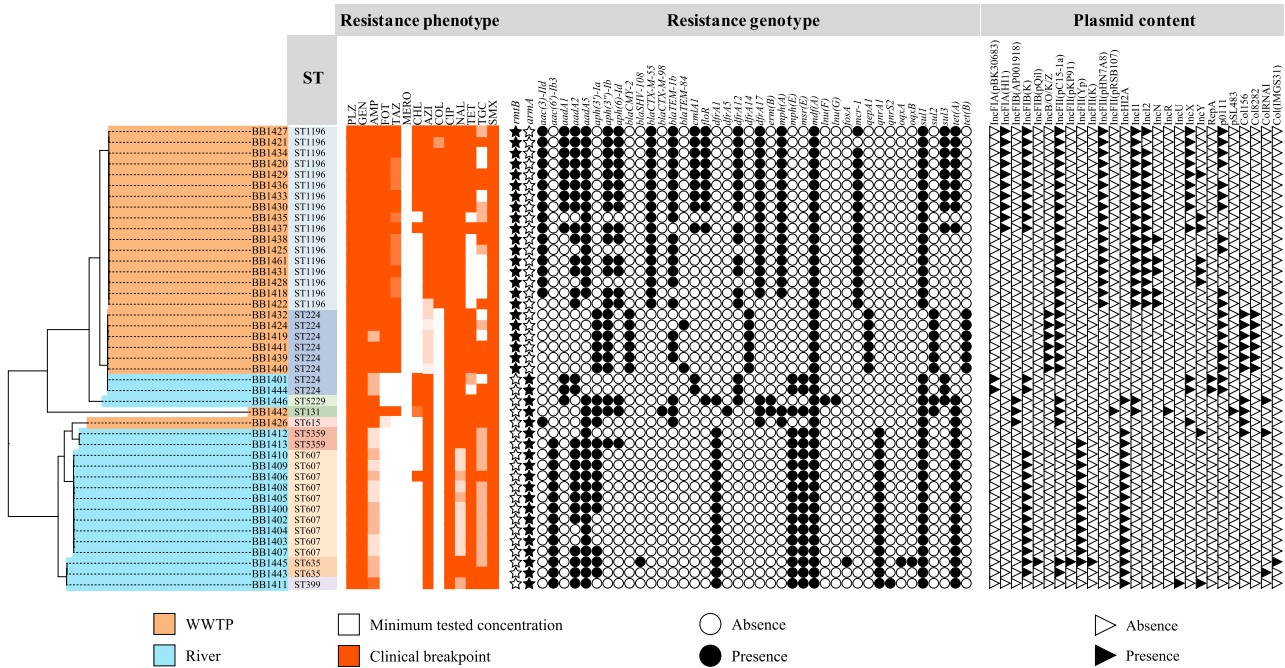

**Fig. 2 Sequenced *E. coli* data.** The source of the isolates is specified by different colors in the genomic SNP-tree branches, as well as the related sequence type. Level of resistance to all tested antibiotics is shown in a gradient of colors: PLZ (plazomicin), GEN (gentamicin), AMP (ampicillin), FOT (cefotaxime), TAZ (ceftazidime), MERO (meropenem), CHL (chloramphenicol), TMP (trimethoprim), AZI (azithromycin), COL (colistin), CIP (ciprofloxacin), NAL (nalidixic acid), TET (tetracycline), TGC (tigecycline), and SMX (sulfamethoxazole). The presence and absence of antibiotic resistance genes and plasmid incompatibility groups are indicated by circle and triangle symbols, respectively. The presence and absence of specific 16S-RMTase genes are indicated by star symbols.

*E. coli* isolates, with the independence of the origin, was constituted by a total of 13,819 different genes. The pan-genome was distributed in a common core-genome of 3109 genes (22.5%) and a variable accessory-genome of 10,710 genes (77.5%), covering a considerable *E. coli* diversity, considering that the estimation of the global *E. coli* core-genome comprises around 1500 genes[19] (Supplementary Fig. 2). Attending to the pan-genome configuration depending on the water source, 3340 out of a total of 9099 genes (36.71%) were included in the core-genome of wastewater isolates, and the core-genome of river isolates was formed by 3410 out of 9927 genes (34.35%), showing that river isolates presented a larger total gene pool and a smaller relative core-genome comparing to WWTP isolates. The genes conforming the pan-genomes of wastewater and river water *E. coli* were statistically different, considering the genes that were present and absent in each niche from the total pan-genome (Jaccard, *P*-value = 0.001) (Fig. 3a), indicating that the different environments, and/or upstream environments from which they have seeded, led to the selection of specific genomic populations, even between members of *E. coli* ST224 originating from the two different environments (Jaccard, *P*-value = 0.049). The genomic diversity of the whole *E. coli* population from river water was significantly higher than the one found in wastewater (Jaccard, *P*-value = $3.599 \times 10^{-8}$) (Supplementary Fig. 3a). However, considering the number of different STs constituting the genomic pool of each environment, the genetic diversity of an *E. coli* ST from wastewater was statistically similar to the diversity of an *E. coli* ST from freshwater (Jaccard, *P*-value = 0.3123) (Fig. 3b). This model, which took into account the number of STs from each source in the diversity analysis, was previously checked by random sampling analysis, obtaining similar results. Essentially, the diversity of the total *E. coli* population was dependent on the number of different STs that defined the population, but the diversity of a specific ST was

independent of this factor and more suitable to estimate the variability of specific bacterial clones according to the origin. Likewise, the diversity of plasmid content, taking into account the different plasmid incompatibility groups identified, was distinctive for each water type (Jaccard, *P*-value = 0.001) (Fig. 3c). Thus, the ecological niche also influenced the presence of specific plasmid types. The level of complexity of the *E. coli* plasmid pool circulating in each aquatic environment was statistically similar (Jaccard, *P*-value = 0.2447) (Supplementary Figure 3b). However, the number of different plasmid types carried by each *E. coli* ST was significantly higher in WWTPs than in rivers (Jaccard, *P*-value = $1.037 \times 10^{-14}$) (Fig. 3d). Both wastewater and river *E. coli* populations also showed a great divergence in the antibiotic resistance gene content (Jaccard, *P*-value = 0.001) (Fig. 3e), showing different resistance mechanisms to particular antimicrobial classes. Furthermore, this diversity was significantly higher in the total *E. coli* population from wastewater comparing to the *E. coli* population from river water (Jaccard, *P*-value = $5.093 \times 10^{-14}$) (Supplementary Fig. 3c), even when the latter possessed a higher total genomic diversity. Thus, the sum of genetic resistance determinants carried by each *E. coli* ST from wastewater environments was much higher than for each *E. coli* ST from natural effluents (Jaccard, *P*-value < $2.2 \times 10^{-16}$) (Fig. 3f).

Regarding the predicted functions associated with the gene clusters from the *E. coli* pan-genome, 2377 different functions were identified in both wastewater and river isolates, of which 1973 (83%) were included in the functional core-genome and 404 (17%) were assigned to the functional accessory-genome (Supplementary Fig. 4). Although the accessory-genome was considerably larger than the core-genome at the gene level, most of the accessory-genome genes were involved in core-genome functions. The total genetic functions encoded in wastewater

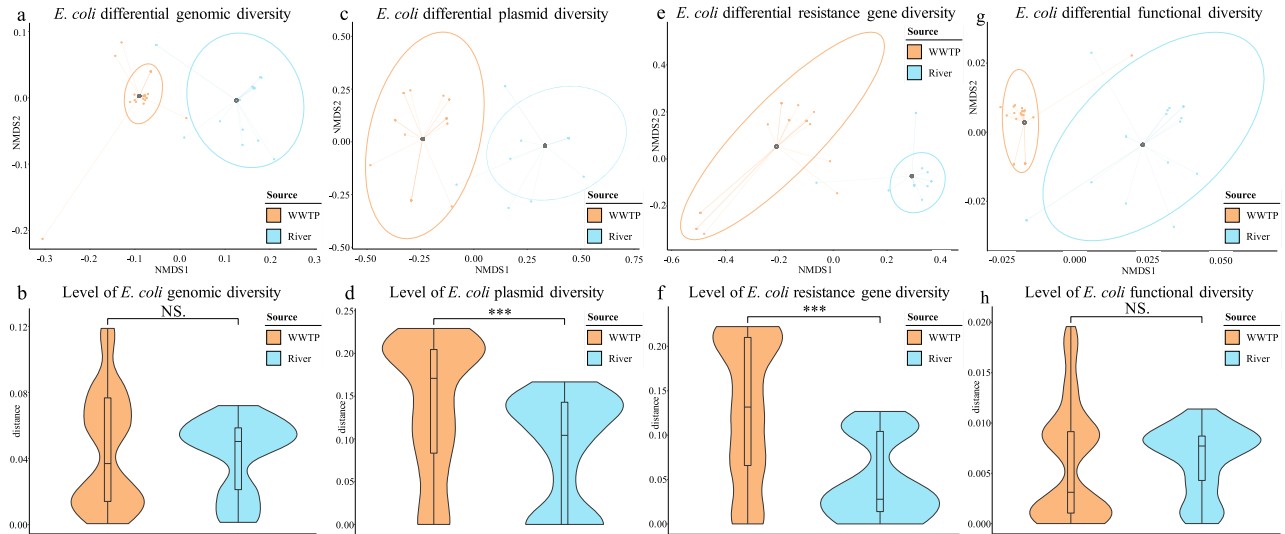

**Fig. 3 Non-metric multidimensional scaling (NMDS) and violin plots according to Jaccard distance index and the environment (WWTP and river, indicated by colors).** NMDS graphs show the differential content in the total wastewater *E. coli* population ($n = 25$ independent isolates) and river *E. coli* population ($n = 18$ independent isolates), indicating the centroids for the means of each group and clustering them by colored ellipses. Violin plots show medians and quartiles for diversity levels per *E. coli* ST, comparing them by the origin of the isolates. Highly significant differences are indicated by ***. Non-significant differences are indicated by NS. **a** Differential genomic content based on core-genome analysis. **b** Level of genomic diversity per *E. coli* ST based on core-genome analysis. **c** Differential plasmid content based on plasmid incompatibility groups. **d** Level of plasmid diversity per *E. coli* ST based on plasmid incompatibility groups. **e** Differential antibiotic resistance gene content. **f** Level of antibiotic resistance gene diversity per *E. coli* ST. **g** Differential function content based on function prediction from core-genome analysis (BB1442 was excluded from NMDS graph to optimize the visualization). **h** Level of function diversity per *E. coli* ST based on function prediction from core-genome analysis.

isolates was 2291, of which 2021 were shared by all of them (88.21%), similar to the 2305 genetic functions identified in river isolates, including the 2023 (87.77%) belonging to the river functional core-genome. Like the pan-genome results at genetic level, the functional diversities found in both water environments were statistically different (Jaccard, $P$-value = 0.001), showing that the functions are closely linked and specific for each ecological niche (Fig. 3g). Due to the greater complexity of the total genomic content, the number of different functions carried out by the entire river *E. coli* population was significantly higher comparing to the wastewater *E. coli* population (Jaccard, $P$-value = $2.137 \times 10^{-12}$) (Supplementary Fig. 3e). Nonetheless, the degree of functional diversity, considering the different STs presented in each environment, was not statistically different (Jaccard, $P$-value = 0.2637) between human-related and natural water *E. coli* (Fig. 3h). Among functions conducted by wastewater isolates, 51 of them were specific from this niche compared to river isolates. These functions were specially related to antibiotic resistance and plasmid processes, which according to the antibiotic resistance and plasmid diversities showed higher levels in WWTPs. On the other hand, 45 functions were well-represented in river isolates but absent in wastewater environments, predominantly including functions linked to regulation processes and metal metabolism (Fig. 4). Interestingly, specific phage functions were representative for each of the ecological niches, suggesting that particular phages were associated with certain environments, as described recently[20].

Long- and short-read hybrid assembly allowed the complete resolution of the genomic structure of 15 representative *E. coli*, attending to the origin, ST, antibiotic resistance gene content, and plasmid content. The length of the chromosomes of *E. coli* from wastewater samples ranged between 4,674,785 bp and 4,979,626 bp, while isolates from rivers possessed chromosomes between 4,695,951 bp and 5,213,442 bp, with most of them being larger than 5 Mb. Besides, the chromosome of isolates belonging to the

same *E. coli* ST from rivers also showed a greater genetic structural variation comparing with the chromosome of wastewater *E. coli* STs, which was highly conserved among their members. Most of the genes involved in this chromosome diversity of river isolates were associated with mobile genetic elements other than plasmids, specially insertion sequences and transposons from multiple families, which, in turn, could act as source of chromosome variation by favoring the integration of other genetic elements. Phage-related genes were also significant elements for chromosome variation of river *E. coli* STs, pointing out the abundance of this kind of elements in natural aquatic environments. Considering antibiotic resistance genes integrated in the chromosome, all isolates from both origins harbored the gene *mdfA* flanked by the same chromosome-related genes and no mobile genetic elements, and no other chromosome resistance mechanism was identified in almost all of the cases. MdfA is a multi-drug transporter commonly found in *E. coli*, which enables resistance to a broad spectrum of chemically different compounds, including antibiotics such as certain aminoglycosides and fluoroquinolones[21]. Among the 15 representative *E. coli* isolates, a total of 70 plasmid structures were assigned to at least one plasmid incompatibility group, of which only 8 (11.43%) showed more than one replicon type in the same structure. The majority of these hybrid plasmids were comprised of replicons belonging to the IncF type family, which are well-known for their capability to coexist in these chimera structures[22]. None of these hybrid plasmids harbored any 16S-RMTase gene, the genetic resistance targets of the present study. Focusing on the implications of these hybrid plasmids in the plasmidome diversity of the whole *E. coli* populations, the diversity analyses were based on the different replicons identified in the isolates, not the independent plasmid structures, since hybrid plasmids are the result of plasmid fusions, reflecting a higher plasmid diversity. Furthermore, the presence of more than one replicon confers a great adaptative advantage to the plasmid structure to be transferred to other bacteria and it can potentially

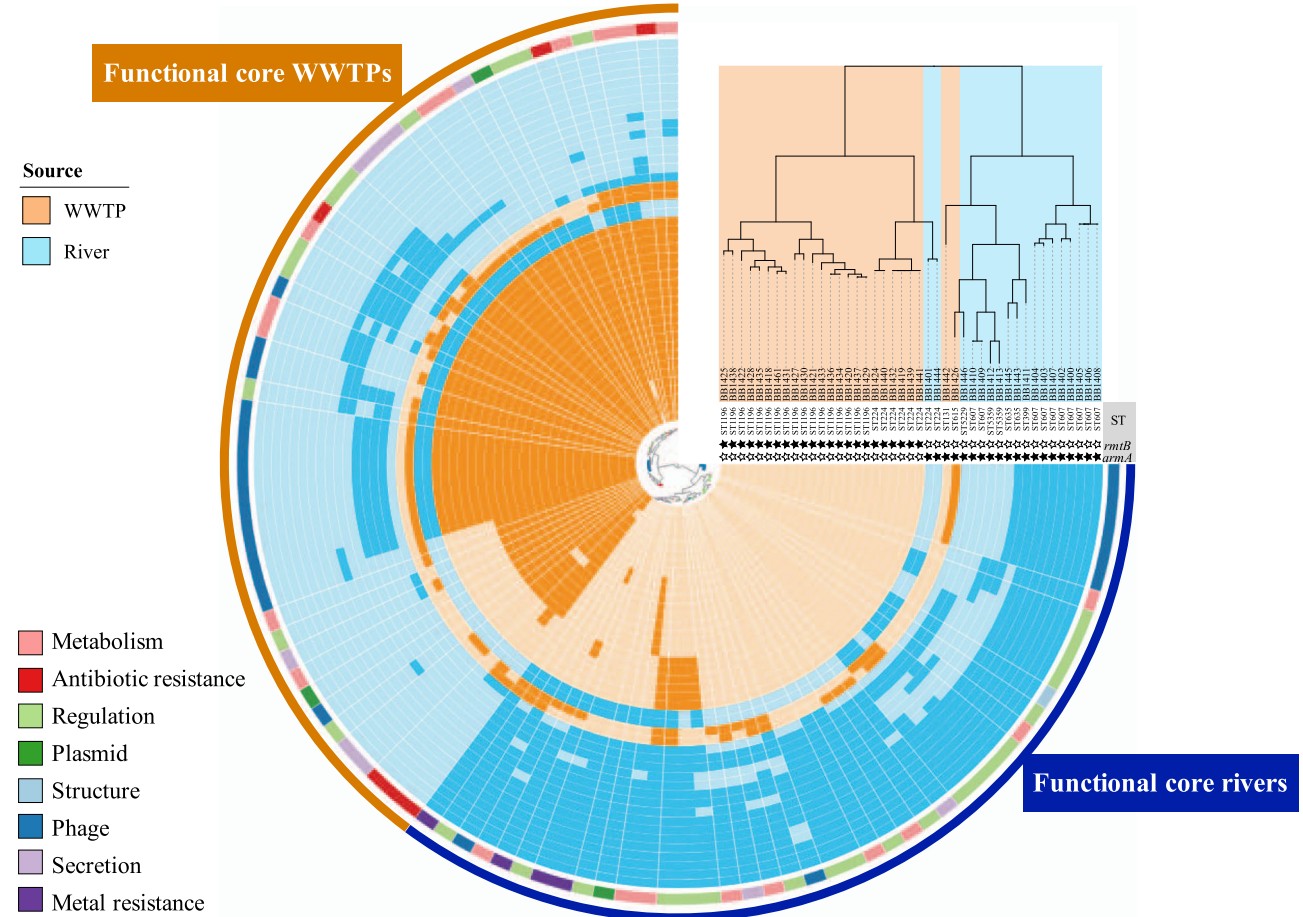

**Fig. 4 Functional core-genome *E. coli* analysis.** The origin of the isolates is indicated by colors in the tree and the circular diagram. *E. coli* ST is indicated in the tips, as well as the presence (black star) and the absence (white star) of 16S-RMTas genes. Functional core-genome tree distribution is based on all predicted functions. A circular diagram shows the presence (dark color) and absence (light color) of functions that are specifically related to one of the environments. Inner ring colors indicate the different functions according to the classification shown in the legend. Outer ring colors indicate the functions related to the functional core-genome of wastewater and river isolates.

split into independent structures[23], which supported the use of the replicons as the key marker to study plasmidome diversity.

**Aminoglycoside resistance is disseminated by distinctive plasmid mechanisms in wastewater and river water.** 16S-RMTase genes were located in plasmids in all wastewater and river isolates. *rmtB* was integrated in an IncFII plasmid type, specifically a pHN7A8-like plasmid, in all isolates sequenced by long-read WGS and belonging to ST1196, which came from wastewater samples. pHN7A8 is a 76,878 bp multi-drug resistance plasmid firstly characterized in China a decade ago and closely involved in *rmtB* dissemination[24]. The pHN7A8-like plasmids identified in ST1196 *E. coli* were 69,320 bp in length and shared 96.0% minimum pairwise nucleotide identity among them and 90.3% with the original plasmid (Fig. 5a). Focusing on the variable region of the sequenced plasmids (the region of the plasmid where genetic rearrangements are focalized because of the presence of mobile genetic elements), the gene content was identical between all pHN7A8-like plasmids, comprising two variants of $bla_{TEM-1}$, $bla_{CTX-M-55}$, and *rmtB*, which were flanked by four copies of IS26 (Fig. 5b). Despite the high conservation of the structure amongst our sequenced plasmids compared to the original pHN7A8 structure, the resistance gene content showed specific differences, such as the loss of *fosA3* and the integration of $bla_{CTX-M-55}$ instead of $bla_{CTX-M-65}$, which could be favored by

the abundant presence of IS26. These divergences were also present at SNP level comparing all the sequenced plasmids around the world with the same pHN7A8 origin of replication (46 plasmids), showing a high similarity among Barcelona plasmids, which were distantly related to all other plasmids. Almost all of them were isolated in Asia, especially China, where this plasmid type was predominantly confined and showed a common structure. Only two plasmids were located out of Asia, in Bolivia, but they were closely related to Asian plasmids with 9 SNPs of difference from the original pHN7A8, demonstrating a possible transcontinental dissemination[25]. However, the pHN7A8 plasmids characterized in WWTPs of Barcelona presented between 90 and 103 SNPs against the Chinese plasmid (Supplementary Fig. 5a, https://microreact.org/project/yeLN26yJf).

Considering the ST224 isolates, they also harbored the *rmtB* gene in an IncFII plasmid type but similar to plasmid pC15-1a. This plasmid, originally described in Canada, was the first $bla_{CTX-M-15}$-carrying plasmid completely sequenced, with a size of 92,353 bp, and it is well-known for its broad geographical distribution and genetic content plasticity[26]. pC15-1a-like plasmid identified in ST224 was 76,521 bp-sized and presented 67% nucleotide identity with the original pC15-1a. Plasmids with this same origin of replication were found in other STs identified in both wastewater, including all ST1196 isolates, and river *E. coli* (Fig. 5a). However, no 16S-RMTase gene was integrated in pC15-1a-like plasmids carried by these other STs, except for the ST615 isolate

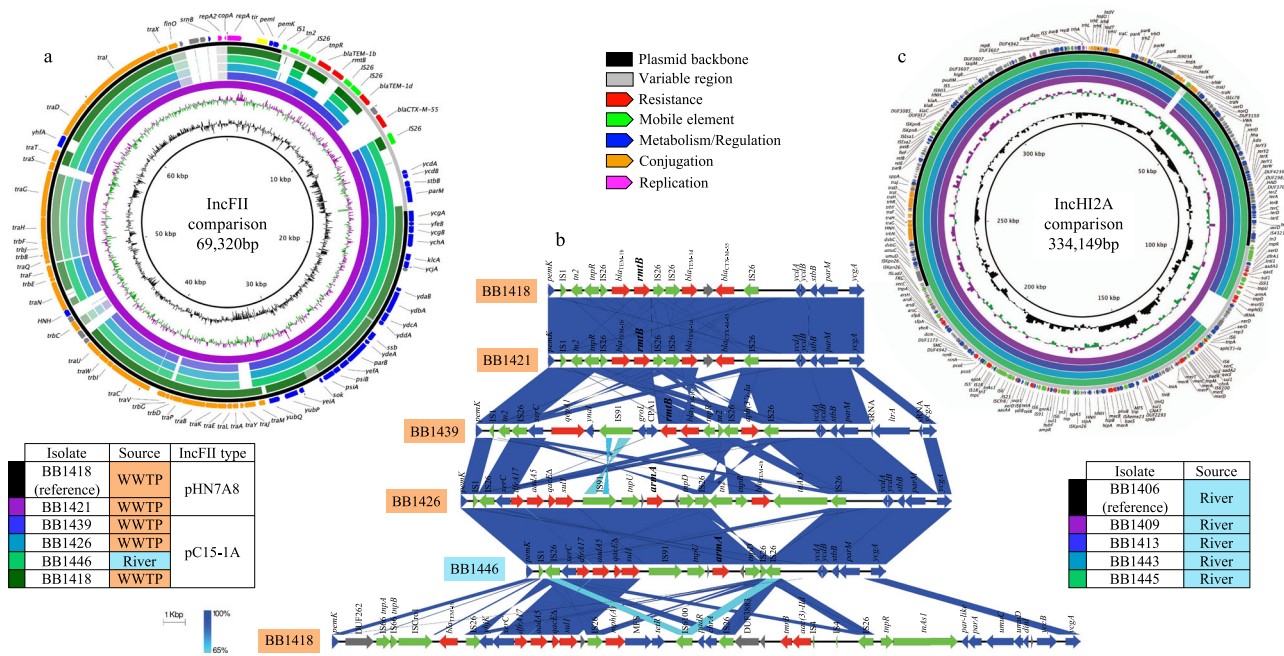

**Fig. 5 Plasmid structure comparison of 16S-RMTase gene-carrying plasmids. a** IncFII type plasmids comparison. From the inside out, the rings represent the reference plasmid (black inner ring), the GC content (black ring), the GC skew (purple/green ring), the compared plasmids (order, carrying isolate, source of the isolate, and IncFII plasmid type are shown in the table), backbone/variable region of the reference plasmid (black/gray ring) and genetic annotation for the reference plasmid (arrows are colored according to the functional groups indicated in the legend). **b** Pairwise comparison of the variable regions of IncFII plasmids. Names of carrying isolate are colored according to the source of the isolate. Gradient blue shades represent the sequence identity. Genetic annotation is represented by arrows and colored according to functional groups (legend). **c** IncHI2A type plasmids comparison. From the inside out, the rings represent the reference plasmid (black inner ring), the GC content (black ring), the GC skew (purple/green ring), the compared plasmids (order, carrying isolate and source of the isolate), backbone/variable region of the reference plasmid (black/gray ring) and genetic annotation for the reference plasmid (arrows are colored according to the functional groups indicated in the legend).

from WWTP water and one ST635 isolate from river water, which harbored the *armA* gene instead of *rmtB* and presented different plasmid sizes. The differences among pC15-1a-like plasmids were caused by the multiple mobile genetic elements integrated in the variable region of the plasmids, especially IS26, and were the cause of the numerous and diverse resistance gene combinations (Fig. 5b). Notably, and despite the different origins, the two *armA*-carrying pC15-1a-like plasmids shared a highly similar variable region, except for the absence of *bla*TEM-1b in the ST635 isolate from river water, which could indicate plasmid transfer between these two niches, and the effect of the environment in the plasmid content shaping (Fig. 5b). According to the SNP-tree analysis, including worldwide plasmids with identical origins of replication (60 plasmids), the pC15-1a plasmid family diversified for a long period of time in multiple branches that disseminated in all continents, mainly associated to *E. coli* recovered from human samples, which explained the low nucleotide similarity between their members. pC15-1a-like plasmids identified in Barcelona waters were located in different cluster branches of the SNP-tree (Supplementary Fig. 5a). In the case of the sole ST131 isolate detected, the 16S-RMTase gene identified was *armA*, contrasting with the majority of wastewater isolates, and the gene was integrated in an IncR type plasmid, a different plasmid type to those responsible in the other STs and previously associated with *armA* dissemination[27].

In contrast to the *E. coli* from wastewater, all isolates from river environments, belonging to four different STs, harbored the *armA* gene encoded in a common IncHI2A type plasmid closely related to R478, except for the aforementioned ST635 isolate and one ST224 isolate that presented this gene integrated in an IncX1 type plasmid. R478 is a 274,762 bp resistance plasmid isolated in the USA in 1969 with a complex conjugative machinery[28]. The

size of the 5 R478-like IncHI2A plasmids identified in river isolates and sequenced by long-read WGS ranged between 321,526 and 334,149 bp, with ~50 kb more than the original plasmid, which resulted in a total nucleotide identity of 70.0–70.6%. However, the minimum pairwise nucleotide identity increased up to 95.1% considering only the backbone of the plasmids, and the nucleotide identity between all sequenced plasmids was 92.0% (Fig. 5c). SNP-tree analysis comprising IncHI2A plasmids isolated worldwide (37 plasmids) confirmed that all members were related, and mostly found in species belonging to genus *Enterobacter* such as the genetically closest one to Barcelona plasmids (47–50 SNPs), isolated in the United States of America (USA) (Supplementary Fig. 5b, https://microreact.org/project/yNTuhk1Jw). However, the R478-like plasmid identified in Barcelona has acquired a large gene content and incorporated it in the variable region of the plasmid, including the *armA* gene together with other multiple resistance genes, due to the abundance of mobile and integrative genetic elements located along the variable region, comprising different IS families and transposons (Fig. 5c). Remarkably, these IncHI2A plasmids also carried numerous resistance determinants to diverse metals, including the sets of genes *ter, mer,* and *ars,* which confer resistance to tellurium, mercury, and arsenic, respectively. These metal resistance mechanisms are commonly associated to IncHI2A plasmids[28] and they were directly linked with the metal metabolism functions specifically identified in river water isolates but not in wastewater ones, since most of the river *E. coli* harbored the IncHI2A plasmid and, therefore, their associated metal resistance determinants. Furthermore, the conjugative machinery of IncHI type plasmids is intimately regulated by *htdA* and *trh* genes, which are expressed in a

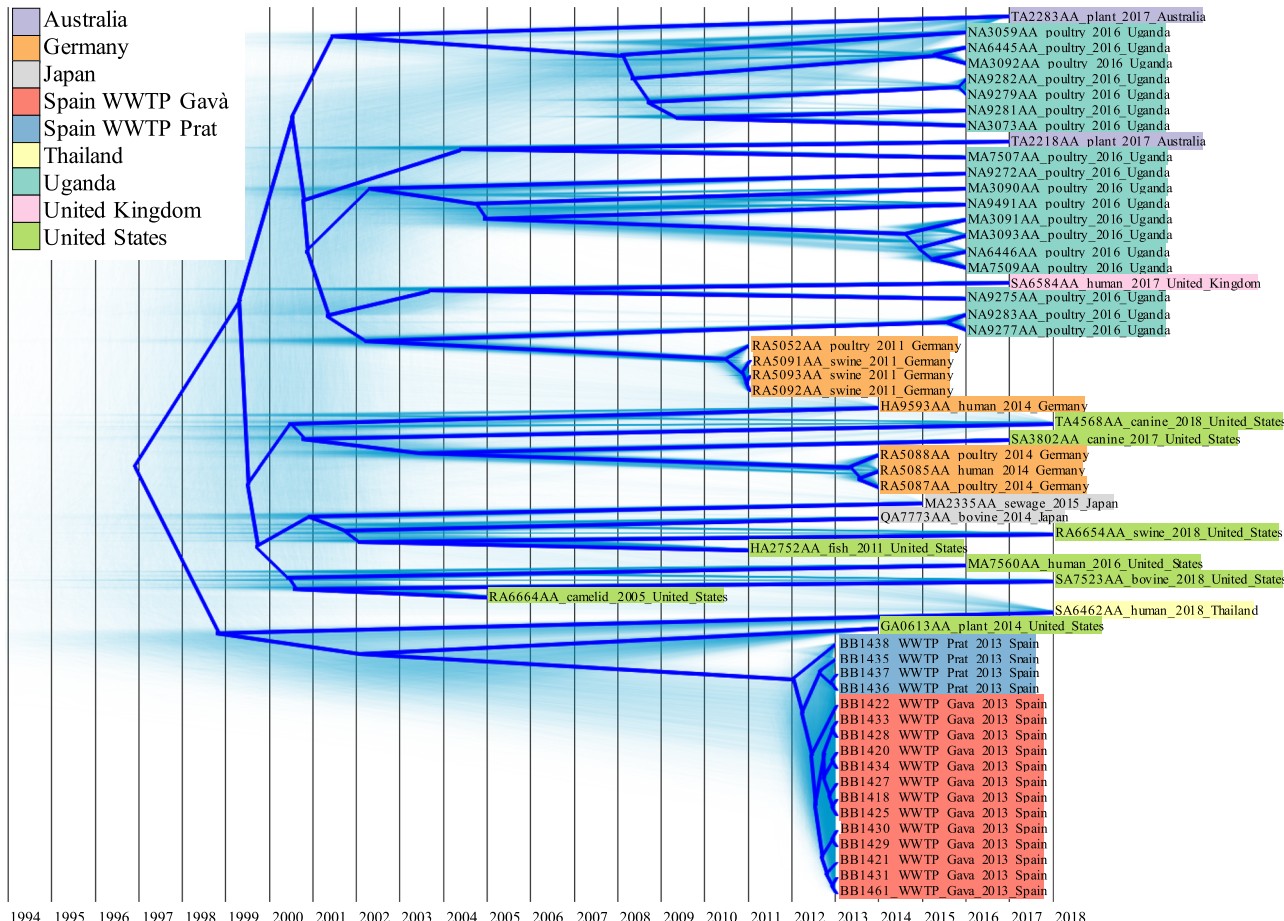

**Fig. 6 Phylogenetic tree of *E. coli* ST1196.** Probable phylogenetic reconstructions are shown with shaded green trees. Summarized root-canal tree is indicated by thick blue tree. Grid indicates the years to locate the divergent events through the evolutionary history of the ST. The name of the isolate, the source, the year of isolation, and the country of origin are indicated in the tips of the tree and colored according to the country of origin as shown in the legend.

temperature-dependent manner. At around 37 °C, *htdA* is expressed, and the resulting protein interacts with Trh factors, transcriptional factors of transfer-related genes, blocking their expression and repressing the conjugation process (Fig. 5c). At the optimal temperature for IncHI conjugation, between 22 and 28 °C, Trh factors are functional, thus the dissemination of this plasmid type has been related to natural water environments[29].

The three plasmid types carriers of 16S-RMTase genes in *E. coli* from aquatic environments of Barcelona (pHN7A8-like, pC15-1a-like, and R478-like plasmids) also harbored the key genetic elements that comprise a complete conjugative machinery: the relaxase or MOB gene (MOBF family for IncFII pHN7A8-like and pC15-1a-like plasmids and MOBH family for IncHI2A R478-like plasmids), the type 4 coupling protein (T4CP) gene and the type 4 secretion system (T4SS) ATPase VirB4 gene, together with 10–12 accessory genes belonging to the type F conjugative system (Figs. 5a, c). These findings confirmed the potential capacity of these plasmids to be transferred to other bacteria via conjugation, allowing the dissemination of their genetic resistance content, including the 16S-RMTase genes.

**Worldwide emergence and dissemination of aminoglycoside-resistant *E. coli* STs are led by different evolutionary events.** SNP-tree analysis of the chromosomes of the three main *E. coli* STs identified in Barcelona waters uncovered its relationship with other isolates belonging to these groups (40 genomes belonging to ST1196, 45 to ST224, and 11 to ST607) and its dissemination in

the world (Supplementary Fig. 6, https://microreact.org/project/a9-1ZIze0). A total of 1322 SNP positions were extracted in non-recombinogenic regions of ST1196 *E. coli* chromosomes, of which between 0 and 5 SNPs were detected among Barcelona isolates and between 72 and 77 SNPs comparing these to the genetically closest isolate. Furthermore, the ST1196 genomic dataset exhibited a high temporal signal (correlation coefficient = 0.8102) and a constant mutational rate (Supplementary Fig. 7a), allowing the phylogeo-graphic reconstruction of this ST. *E. coli* ST1196 was estimated to have emerged at the end of 1996, and the dissemination focus was located in Germany, from where this ST spread worldwide in a short period of time, especially to the USA at the beginning of 2000, which was the possible country of origin of the isolates identified in Barcelona WWTPs (Fig. 6) (See Supplementary Movie 1 and Supplementary Data 2 to visualize an interactive map in Google Earth website[30] by importing it as a project). We found no evidence of the spread of ST1196 from Barcelona to new locations, but new routes of dissemination from Germany, the USA, and Uganda to other countries were identified in the last two years. Most of the ST1196 *E. coli* have been sequenced from animal sources, mainly from poultry samples in Germany and Uganda, although the most closely related isolate to those from the Barcelona WWTPs was isolated from a plant in Florida (USA).

The ST224 *E. coli* chromosomes accumulated a total of 7246 SNPs along their non-recombinogenic regions. This divergence was mainly due to the existence of two clusters genetically quite distant (Supplementary Figure 6, https://microreact.org/project/a9-1ZIze0). Furthermore, the members belonging to the same

cluster also presented numerous SNPs. In fact, Barcelona ST224 isolates from wastewater and river water were well-differentiated by 92–95 SNPs, indicating a possible divergent evolution in each environment from a common ancestor or an unrelated origin between them. However, WWTP isolates shared an identical chromosome (0 SNPs), whereas river isolates presented 6 SNPs between them. Focusing on the SNP analysis results of ST607 *E. coli*, the total SNPs detected in the chromosome dataset were 1713, but Barcelona river isolates were distributed in different clusters genetically separated by 177–181 SNPs, which highlighted the higher chromosomic variation exhibited by river isolates comparing to WWTP isolates (Supplementary Figure 6, https://microreact.org/project/a9-1ZIze0). Due to the heterogeneity among *E. coli* isolates, temporal signal and mutational rate of ST224 (correlation coefficient=0.1644, Supplementary Fig. 7b) and ST607 (correlation coefficient = 0.3548, Supplementary Fig. 7c) were not suitable to carry out posterior molecular clock analysis. Remarkably, no 16S-RMTase gene has been previously reported to be harbored by any of these three STs (ST1196, ST224, and ST607) (Supplementary Fig. 6, https://microreact.org/project/a9-1ZIze0).

## Discussion

The intestinal tract of warm-blooded animals, including humans, has long been considered to be the ecological niche of *E. coli*. This species is therefore thought to be an indicator of fecal contamination[31]. Here we found that this bacterium is the predominant aminoglycoside-resistant *Enterobacteriaceae* identified in WWTPs of Barcelona, but also highly represented in river water samples. Indeed, other studies have demonstrated the existence of indigenous *E. coli* populations inhabiting natural water environments, questioning the confinement of this species to a limited niche spectrum[31,32]. The human or animal-related origin of river *E. coli* isolates cannot be totally excluded. However, the environmental conditions of the sampled rivers and the significant differences found between wastewater *E. coli*, originated from the human and animal populations of the region, and river *E. coli* pointed at the natural likely origin of the latter ones. *K. pneumoniae* was the predominant aminoglycoside-resistant *Enterobacteriaceae* in rivers, but its prevalence in WWTPs was considerably lower than that of aminoglycoside-resistant *E. coli*. While all isolates were highly resistant to all clinically relevant aminoglycosides, even last resort aminoglycoside plazomicin, since this resistance profile was the target of the study, the responsible gene was clearly segregated according to the environment, with the *rmtB* gene in wastewater and *armA* in rivers. This could explain the different prevalence of aminoglycoside-resistant *Enterobacteriaceae* species between the two environments, since *rmtB*, the predominant 16S-RMTase gene in WWTPs, and their carrying mobile genetic elements are more frequently associated with *E. coli* comparing with *K. pneumoniae*[33]. Likewise, we found differences in the genomic diversity, plasmid content, and resistome of *E. coli* from human-related and natural water samples, which likely reflects both the incoming sources of these different populations as well as the environmental conditions to which they are subsequently exposed. As a consequence, wastewater and river water *E. coli* presented specific genetic compositions, but the level of diversity according to the ST variety was similar in both environments, revealing that WWTP *E. coli* harbored a richness greater than thought, and even equivalent to that found in *E. coli* from natural ecosystems. Furthermore, the level of functional diversity in wastewater *E. coli* STs was similar to the level detected in river *E. coli* STs, demonstrating that the genetic richness of both environments was translated into a comparable functional

richness. The *E. coli* genomic diversity in WWTPs was greatly due to the higher plasmid diversity circulating, whereas the genomic diversity in river *E. coli* was caused by variations in their larger chromosomic structures, including the integration of mobile genetic elements. Hence, and taking into consideration that numerous antibiotic resistance genes are usually encoded in plasmid structures[34], the larger plasmid diversity of WWTP isolates was one of the main factors contributing to the higher antibiotic resistance gene diversity found in wastewater environments. Consequently, several antibiotic resistance functions were characterized as indicators for a human-related origin.

In WWTP environments, *rmtB* was integrated in diverse plasmids, each of which was associated with a specific *E. coli* ST. ST1196 isolates carried the gene in a pHN7A8-like plasmid, a plasmid type with a high level of conservation in its structure through space and time. However, those from WWTPs of Barcelona were the most genetically distant to all plasmids belonging to the same replicon type, revealing its establishment and divergent evolution in Europe. The *rmtB*-pHN7A8-ST1196 association formed in Barcelona WWTPs was highly prevalent and their members showed a close relationship among them, being a potential risk for public health. A pC15-1a plasmid was responsible for *rmtB* dissemination in ST224 *E. coli* and was also identified in other STs. Due to its promiscuity and capability to capture and rearrange different resistance genes, the acquisition of pC15-1a plasmid is considered a trigger for epidemic clones worldwide[26]. This is the case of ST224 *E. coli*, which presented high genetic plasticity that has allowed its dissemination in all continents and its adaptation to all kinds of ecological niches, being present in WWTPs and rivers of Barcelona. The variety of pC15-1a-like plasmids identified in Barcelona suggested either a multi-dissemination from different origins that have converged in this region or a divergent evolution from a common ancestor. Interestingly, the two IncFII type plasmids identified, pHN7A8 and pC15-1a, have a common progenitor, R100 plasmid, whose structure has undergone serial evolutionary events that led to the emergence of these and other various clinically relevant IncFII plasmids[24]. On the contrary, *armA* gene was integrated in a R478-like plasmid in river isolates, independent of the *E. coli* ST. The backbone structure of this plasmid was highly preserved amongst Barcelona river isolates compared to the plasmid isolated 50 years ago[28], and its variable genetic content was shared by all Barcelona plasmids. However, isolates belonging to the ST607 *E. coli*, the predominant ST in river environments, were genetically distant, demonstrating the higher chromosomic diversity found in natural water environments. In a global health context, we have described the association of three epidemiologically relevant *E. coli* STs (ST1196, ST224, and ST607) with aminoglycoside pan-resistance mechanisms. Overall, wastewater environments promoted the expansion of conserved *E. coli* STs and the flux of resistance genes via plasmid structural variation and dissemination, resulting in specific plasmid-ST associations. This could be due to the introduction and existence of numerous resistance plasmids and bacteria acting as reservoirs and the continuous change of environmental conditions in WWTPs, which allowed the exchange of a diverse genetic repertoire. On the contrary, chromosomic variations were the key for *E. coli* diversification and preservation in river environments, showing a general maintenance of the gene-plasmid associations, which could be favored by a lower anthropogenic impact and constant environmental conditions that selected a more stable bacterial population.

## Methods

**Sampling and bacterial isolation**. Samples were collected from July to November 2013 from river water and wastewater in the region of Barcelona (Spain). A total of eight samples were obtained from two rivers, the Llobregat and the Cardener,

considering the proximity to the sampled WWTPs. The Llobregat river represents one of the principal water sources of the city of Barcelona, and the Cardener river is a tributary of Llobregat river. The Llobregat river receives a low anthropogenic impact in the higher course and the occurrence of antibiotic resistance genes of clinical relevance in both water and sediments is considerable[35]. From Llobregat river, samples were taken at two locations: from surface water (three samples on different dates from location A and one sample from location B) and sediments (one sample from each location). From Cardener river, two samples were collected, one from surface water and one from river sediments. For analysis, 10 mL of river water was filtered through a cellulose membrane (0.45 μm pore size), after which the filter was placed on Mac-Conkey agar plates (Oxoid Ltd., Basingstoke, Hampshire, UK) supplemented with 200 mg/L gentamicin and 200 mg/L amikacin (Sigma-Aldrich Inc., Saint Louis, Missouri, USA)[36]. Likewise, 5 g of sediment from each river was homogenized 10-fold diluted in PBS, centrifuged for 5 min at 300 g and the resulting supernatant was then plated onto MacConkey agar with aminoglycoside pressure (same as surface water samples). Five wastewater samples of 10 mL were obtained from two WWTPs located in Baix Llobregat (the area where the Llobregat river flows into the Mediterranean). Gavà WWTP, serving an area consisting of a number of cities and towns in the area, with a population of 370,000 inhabitants, and El Prat WWTP, collecting the sewage from the south of Barcelona city, corresponding to 2,000,000 inhabitants. Three samples were obtained from El Prat WWTP and two samples were obtained from Gavà WWTP, from different water tanks and on different dates. Samples were 10-fold diluted in PBS and plated onto MacConkey agar supplemented with aminoglycosides (same as above). All lactose-positive colonies were analyzed.

**Bacterial identification, antibiotic susceptibility testing, and detection of 16S-RMTase genes by PCR.** Bacterial species were identified via MALDI-TOF mass spectrometry in Centro de Vigilancia Sanitaria Veterinaria (VISAVET, Madrid, Spain). Briefly, samples were obtained by ethanol formic acid extraction from fresh and pure cultures and merged with α-cyano-4-hydroxy-cinnamic acid (HCCA) matrix. Data acquisition was performed using a Bruker Daltonics UltrafleXtrem MALDI TOF/TOF equipment and the Biotyper Real-Time Classification software version 3.1 with the MALDI Biotyper database (Bruker Daltonics, Bremen, Germany). Resistance levels to different antibiotic classes (penicillins, cephalosporins, carbapenems, monobactams, fluoroquinolones, tetracyclines, fosfomycin, trimethoprim-sulfamethoxazole, and aminoglycosides) (Bio-Rad Laboratories Inc., Hercules, California, USA) were initially evaluated by disk diffusion method following the European Committee on Antimicrobial Susceptibility Testing (EUCAST) guidelines[37]. When the information was not available in these guidelines, Clinical and Laboratory Standard Institute (CLSI) guidelines[38] were used. PCRs for all known 16S-RMTase genes (*armA*, *rmtA*, *rmtB*, *rmtC*, *rmtD*, *rmtE*, *rmtF*, *rmtG*, *rmtH*, and *npmA*) were performed as previously established, using defined primers and conditions[39].

Subsequently, MIC to different antibiotics of selected *E. coli* isolates was assessed by broth microdilution method using commercial Sensititre EUVSEC plates (Trek Diagnostics Inc., Westlake, Ohio, USA) following the manufacturer's specifications and the aforementioned guidelines for the interpretation. The origins, antibiotic resistance profiles, and genetic relatedness of isolates were used in the selection of isolates for MIC evaluation, which aimed to maximize the diversity of those chosen.

**Clonal relatedness.** Clonality profiles of all 16S-RMTase-positive *E. coli* were determined by Pulsed Field Gel Electrophoresis (PFGE) as previously established[36]. Briefly, DNA plugs were digested using XbaI restriction enzyme (Takara Bio Inc., Otsu, Shiga, Japan) for 14 h. PFGE was undertaken on a CHEF-DR III (Bio-Rad Laboratories Inc.) using the following parameters: running time 22 h, temperature 14 °C, field strength 6 V/cm², angles 120º, initial pulse time 2.2 s, final pulse time 63.8 s. The dendrogram was constructed with XbaI-PFGE profiles using Bionumerics (Applied Maths N. V., Sint-Martens-Latem, Belgium) version 6.0 (UPGMA cluster analyses based on Dice correlation coefficient, tolerance of 2%, and optimization of 1%). A cutoff similarity value ≥90% was used to define clones. Isolates with ambiguous PFGE profile were not included in the dendrogram analysis.

**Total DNA extraction.** Genomic DNA extraction and purification of selected 16S-RMTase-positive *E. coli* was performed using Wizard Genomic DNA Purification Kit (Promega Corp., Madison, Wisconsin, USA), following the Isolating Genomic DNA from Gram-Negative Bacteria protocol. Afterwards, DNA quality and concentration were measured by NanoDrop (Thermo Fisher Inc., Waltham, Massachusetts, USA) and Qubit (Invitrogen Corp., Carlsbad, California, USA) devices. The selection of *E. coli* isolates for DNA extraction and subsequent whole-genome sequencing was conducted according to their origin, antibiotic resistance profile, and genetic relatedness in order to include all clonal diversity.

**Whole-genome sequencing by Illumina and data processing.** High-throughput genome sequencing was carried out in the Instituto de Biomedicina y Biotecnología de Cantabria (IBBTEC, Santander, Cantabria, Spain). Shortly, 101 bp paired-end reads (550 bp insert size) were generated on a HiSeq 2500 platform (Illumina Inc., San Diego, California, USA), obtaining a mean of 141-fold coverage (minimum 88-fold, maximum 234-fold). Short-read sequences were processed for subsequent

analysis by checking its sequencing quality with FastQC version 0.11.3[40] and trimming sequencing adapters and end nucleotides with low quality using Trimmomatic version 0.33[41]. Raw Illumina sequence data were deposited under the project PRJEB34557 in the European Nucleotide Archive (ENA)[42], and individual accession numbers are indicated in Supplementary Data 3.

**Short-read de novo assembly and annotation.** De novo assembly of each sequenced isolate was obtained using SPAdes version 3.11.0[43], using paired-end reads and k-mer sizes ranging from 57 to 117. Assembly qualities were assessed with QUAST version 4.6.0[44], with N50 values varying between 68282 and 347758. All contigs from each assembly were annotated via Prokka version 1.5[45]. Short-read assembly data were submitted to ENA[42] under the project PRJEB34557, and genome accession numbers are indicated in Supplementary Data 4.

**Sequence type, resistome, and plasmidome analysis.** ST of sequenced *E. coli* was determined through MLST using the MLSTcheck tool version 1.007[46] for *E. coli* including Pasteur and Warwick MLST schemes[47], extracting allele sequences from de novo assemblies. Antibiotic resistance genes and plasmid incompatibility groups present in each of the isolates were identified using ARIBA pipeline version 2.12.1[48] by mapping processed short-reads against ResFinder[49] and PlasmidFinder[50] databases, respectively. Origin, clonal relatedness, ST, phenotypic resistance, antibiotic resistance genes, and plasmid content results were visualized with iTOL version 4.4.2[51].

**Pan-, core- and accessory-genome analysis.** Pan-genome study was carried out following two different pipelines. Firstly, all annotated assemblies from Prokka were used as input to Roary version 3.11.2[52], obtaining the gene families belonging to both accessory- and core- genome. SNP (single nucleotide polymorphism) extraction was subsequently performed from core-genome alignment using SNP-sites tool version 2.4.1[53], and the variable positions in the core-genome were used to construct a core-genome SNP-tree via RAxML version 8.2.8[54] with 100 bootstrap replicates. FigTree version 1.4.3[55] was employed to visualize the SNP-tree. This core-genome SNP-tree was visualized together with the gene-presence absence matrix generated by Roary using Phandango version 1.3.0[56]. Secondly, we used anvi'o version 5.5[57] applying the workflow for microbial pangenomics[58]. Briefly, anvi'o databases were generated for all assemblies using the program 'anvi-gen-contigs-database', including recognition of ORFs in contigs via Prodigal version 2.6.3[59]. Subsequently, the program 'anvi-run-ncbi-cogs' was run to identify amino acid gene sequences from the NCBI's Cluster of Orthologous Groups (COG) database[60]. Then, pangenomic analysis was carried out using the program 'anvi-pan-genome' with the annotated contig databases and default parameters, and a categorization to differentiate river and WWTP isolates was added with 'anvi-import-misc-data'. In addition, average nucleotide identity (ANI) was computed across all genomes with 'anvi-compute-ani' program using PyANI version 0.2.7[61]. The analysis results were graphically display on the anvi'o interactive interface with the program 'anvi-display-pan', organizing the genomes by gene clustering and editing the figure to specify the origin of the isolates and the accessory- and core-genome. Afterwards, functions were assigned to all annotated gene clusters and enriched functions taking into account the origin of the isolates (river or WWTP) were identified via 'anvi-get-enriched-functions-per-pan-group' program. Resulting functional analysis was filtered in order to preserve functions with a corrected p value for the enrichment score of 0, meaning those functions that were present in a high proportion of isolates from river or WWTP and belonged to the functional core-genome of one of these two groups, differentiating the isolates according to its origin. These functions were manually curated and classified in different functional groups. Finally, presence-absence analysis of all annotated functions was used to generate a functional pangenome tree with 'anvi-matrix-to-newick' program, which was included, together with functional core-genome data, in the final analysis and visualization with 'anvi-interactive' program.

**Bacteria, plasmid, resistance gene, and function diversity analyses.** *E. coli* diversity was studied according to the origin of the isolates, wastewater environments ($n = 25$ independent isolates), and river environments ($n = 18$ independent isolates), in order to detect possible differences in these populations. Analysis was carried out using vegan package version 2.5-5[62] with R version 3.6.1[63] and RStudio version 1.2.1335[64]. Firstly, similarity coefficient was calculated by Jaccard Index for gene presence-absence results from pan-genome analysis and ordinated in non-metric multidimensional scaling (NMDS), specifying the origin of the isolates and estimating the centroids for the means of each group. To assess statistical differences between WWTP and river diversities, permutational multivariate analysis of variance (PERMANOVA) was performed with similarity coefficient values using adonis. Then, values from the distance matrix were extracted for both sample origins and combined to evaluate differences between *E. coli* diversity levels in WWTPs and rivers applying T-test analysis. The data from the Jaccard distance matrix were also extracted and weighed according to the number of different STs present in each origin (four in WWTPs and six in river environments) in order to assess the statistical differences of genomic diversity per *E. coli* ST depending on the origin. This model was previously checked by random sampling analysis, making equal the number of STs per source and the number of members per ST,

obtaining similar results. Distributions of total *E. coli* population diversity and diversity per *E. coli* ST according to the origin were graphically represented through violin plots. Plasmid, resistance gene, and function diversity analysis were developed following the aforementioned process for bacteria diversity analysis, using presence-absence data from plasmidome (plasmid incompatibility groups), resistome (antibiotic resistance gene content), and function (functions assigned to gene clusters of the pangenome) analysis, respectively. The R codes designed and applied for all aforementioned analyses are available at the GitHub (https://github.com/JoseFranciscoDelgadoBlas/bacterial_population_diversity) and Zenodo (https://doi.org/10.5281/zenodo.4542907)[65] repositories.

**Whole-genome sequencing by Nanopore and data processing**. Selected *E. coli* isolates were sequenced using the MinION device (Oxford Nanopore Technologies Ltd., Oxford Science Park, Oxford, UK). DNA genomic extraction was performed as described above, avoiding DNA fragmentation as far as possible. Genomic libraries were performed following the 1D Native barcoding genomic DNA protocol, with EXP-NBD103 and SQK-LSK108 kits (Oxford Nanopore Technologies Ltd.), and sequencing was run in a FLO-MIN106 flow cell. Downstream analysis was performed as follows: sequencing reads were base called with MinKNOW software (Oxford Nanopore Technologies Ltd.), demultiplexing process was carried out with the Fastq Barcoding workflow of Epi2Me interface (Metrichor Ltd., Oxford Science Park, Oxford, UK) and trimming of adaptors and barcodes from the reads was assessed by Porechop version 0.2.3[66], obtaining a mean of 54.6-fold coverage (minimum 8.5-fold, maximum 113.5-fold). Raw Nanopore sequence data were deposited in the ENA[42] under the project PRJEB34259, and individual accession numbers are indicated in Supplementary Data 5.

**Long-short read hybrid assembly and genomic analysis**. Hybrid assemblies for *E. coli* isolates with short and long sequencing reads were achieved using Unicycler version 0.4.0[67] and applying default parameters, obtaining N50 values between 3673505 and 5213442. Closed genomic structures were assessed by Bandage version 0.8.1[68] and annotated by Prokka. These structures were extracted independent from each assembly, a set of artificial short paired-end reads (100 bp length and 350 bp insertion size) was generated from each of them and analyzed via ARIBA pipeline (as described in the "Sequence type, resistome, and plasmidome analysis" section) with ResFinder and PlasmidFinder databases in order to identify resistance genes and plasmid incompatibility groups, respectively. Long- short-read hybrid assembly data were submitted to ENA[42] under the project PRJEB34259, and genome accession numbers are indicated in Supplementary Data 6.

**Structural analysis of bacterial chromosomes**. Closed and annotated chromosomes of the most prevalent *E. coli* ST identified, in both wastewater and river environments, were aligned by Progressive Mauve version 2.3.1[69]. The resulting chromosome comparisons were graphically visualized using Geneious version 8.1.9[70] and analyzed in order to identify the common and variable chromosome genetic content and structure.

**16S-RMTase gene-carrying plasmid analysis**. Closed structures of most prevalent plasmid incompatibility groups harboring 16S-RMTase genes from the present study were aligned using the 'dnadiff' module version 1.3 for nucmer[71], obtaining the percentage of identity between them at nucleotide level. Then, plasmids of the same incompatibility group were graphically compared with BLAST Ring Image Generator (BRIG) version 0.95[72], taking one of the plasmids as a reference for each group. Variable regions identified by BRIG comparison were extracted from each sequence, manually checked for accurate annotation and aligned using EasyFig version 2.2.2[73], in order to determine and visualize the gene content of these regions. Protein plasmid sequences annotated with Prokka were used to identify and classify the genetic elements associated with conjugative systems by using the program MacSyFinder version 1.0.5[74] and applying the CONJscan models[75].

**Other bacteria and plasmid data included in posterior analysis**. Data mining was carried out to collect *E. coli* and plasmid sequences from other studies worldwide related to those responsible for 16S-RMTase gene dissemination in the present work. In the case of *E. coli* sequences, the search was performed through EnteroBase database version 1.1.2[76], identifying the assemblies of all *E. coli* belonging to the most prevalent sequence types found in this study and with accurate linked metadata (country and year of isolation and sample source) (Supplementary Data 7). Plasmids from particular incompatibility groups obtained from any species were identified by searching for sequences with a 100% identical origin of replication in the National Center for Biotechnology Information (NCBI) database[77]. Afterwards, incomplete sequences and sequences lacking metadata (country and year of isolation, sample source, and bacterial species carrier) were filtered out, leaving only closed plasmids with all required metadata (Supplementary Data 8). A set of artificial 100 bp paired-end reads was generated for each bacteria and plasmid assembly to perform subsequent analysis. Detection of resistance gene content for *E. coli* and plasmid collections, focusing on

16S-RMTase genes, and plasmid content in the case of the *E. coli* strain collection were carried out via the ARIBA pipeline, as described above.

**Read alignment, SNP detection, and SNP-tree analysis of 16S-RMTase-carrying *E. coli* and plasmids**. Short reads of all isolates from the most prevalent *E. coli* STs found in this study and short reads of these STs from other studies worldwide were mapped against one of the closed chromosomes as a reference using Sequence Mapping and Alignment Tool (SMALT) version 0.7.4[78]. SNPs were then extracted from the resulting alignment with SNP-sites and used to obtain a SNP-tree by RAxML, which was visualized through FigTree, as previously specified in Pan- and core-genome analysis section. In addition, the final SNP-tree was associated with isolates' metadata and graphically represented using the interactive tool Microreact[79]. For plasmid SNP-trees, mapping of short reads from the present study and other studies belonging to the predominant 16S-RMTase gene-carrying plasmids was performed against the corresponding plasmid closed reference, following the aforementioned process for *E. coli* SNP-tree construction and establishing a minimum mapped length of the reference of 50% to be included in the SNP-tree.

**Temporal analysis**. Independent mappings were performed for each of the main *E. coli* STs using SMALT, including sequencing data from this study and others and selecting one closed chromosome as a reference for each ST. Then, recombination regions were identified and removed from mapping alignments by Gubbins version 1.4.10[80]. SNPs from the resulting filtered alignments were used to construct a SNP-tree for each ST with RAxML. SNP-trees, together with the year of collection of *E. coli* isolates, were assessed by TempEst version 1.5.1[81] in order to evaluate the temporal signal within STs for posterior molecular clock analysis, proceeding only with datasets with a correlation coefficient ≥0.8.

**Phylogeographic analysis**. Discrete phylogeographic analysis of *E. coli* STs, comprising data from the present study and other studies worldwide, was carried out using Bayesian Evolutionary Analysis Sampling Trees (BEAST) version 2.5.1[82,83]. Datasets with a strong temporal signal were processed using Bayesian Evolutionary Analysis Utility (BEAUti), testing different model combinations. The best-fit model and parameters used were as follows: a generalized time-reversible nucleotide substitution model (GTR), with four relative rates of mutation and 0.0% of invariant sites, applying a strict molecular clock, estimating a uniform evolutionary rate and a constant population size. The sample location of each isolate was added to the model as a discrete trait. The run was performed using 300,000,000 Monte Carlo Markov Chain (MCMC) iterations and 10% burn-in iterations. BEAST results were inspected by Tracer version 1.7.1[84], defining a robust analysis when effective sample sizes (ESSs) for all considered parameters were >200, as recommended by BEAST guidelines. The set of trees produced by BEAST was visualized by DensiTree version 2.2.6[85] and edited to show a root-canal tree, a time-scale, and the country of origin by colors. Then, a summary tree was generated by TreeAnnotator, selecting maximum clade credibility tree and mean values for node heights, and evaluated using FigTree. Afterwards, the summary tree was processed with Spatial Phylogenetic Reconstruction of Evolutionary Dynamics (SPREAD) version 1.0.7[86] to generate a spatio-temporal diffusion analysis using the sample location data, obtaining a.kml file. Finally, the resulting evolutionary events of *E. coli* ST were graphically visualized by uploading the KML file to Google Earth Pro version 7.3.2.5776[30].

**Reporting summary**. Further information on research design is available in the Nature Research Reporting Summary linked to this article.

## Data availability

All raw and assembled sequence data from this study have been deposited in the European Nucleotide Archive (ENA)[42] at EMBL-EBI under the umbrella project PRJEB34801, and properly specified within the paper and its Supplementary Data files. All accession codes for Illumina raw sequence data and assemblies are indicated in Supplementary Data 3 and 4, respectively. All accession codes for Nanopore raw sequence data and hybrid assemblies are indicated in Supplementary Data 5 and 6, respectively. The authors declare that sequence data from public repositories analyzed during the current study are available within the paper and indicated in Supplementary Data 7 and 8. All other relevant data are available from the corresponding authors.

## Code availability

The R codes for bacteria, plasmid, resistance gene and function diversity analyses were designed using vegan package version 2.5-5 with R version 3.6.1 and RStudio version 1.2.1335, as specified in the Methods section. The codes are publicly available at the GitHub (https://github.com/JoseFranciscoDelgadoBlas/bacterial_population_diversity) and Zenodo (https://doi.org/10.5281/zenodo.4542907)[65] repositories.

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

## Acknowledgements

The authors acknowledge Isabel Cuesta and Sara Monzon from the Bioinformatics Unit at the Institute of Health Carlos III for their contributions and advice to the study. The authors thank Almudena Casamayor from the Microbial Identification and Characterization Unit at the VISAVET Health Surveillance Centre for her support in the bacterial identification by MALDI-TOF mass spectrometry. The authors acknowledge Iciar Rodriguez-Avial from the Microbiology Unit at the San Carlos Hospital for providing the plazomicin to perform MIC evaluations. Work carried out in the Institute of Biomedicine and Biotechnology (IBBTEC) was funded by the Ministry of Economy and Competitiveness (grant BFU2017-86378-P). The work was supported by the Spanish Ministry of Economy and Competitiveness (MINECO BES-2015-073164) and the European Union's Horizon 2020 Research and Innovation Programme (grant 773830, OH-EJP-H2020-JRP-AMR-2-WORLDCOM).

## Author contributions

J.F.D.B., C.M.O. and B.G.Z. designed the investigation. W.C.C. and M.M. collected the samples and performed the preliminary screening for pan-aminoglycoside-resistant bacteria. J.F.D.B., C.M.O. and N.M. performed bacterial identification, antimicrobial susceptibility testing and PFGE. M.P.G.B. and F. de la C. performed Illumina sequencing and preliminary analyses of the data. J.F.D.B. and N.M. performed Nanopore sequencing. J.F.D.B. and S.D. designed and performed genomic analysis, including genome assemblies, *E. coli* MLST, antibiotic resistance gene and plasmid analyses, pan-genome analyses, SNP-tree analysis and phylogeographic reconstructions. D.M.A. supervised genomic analysis. B.G.Z. supervised the entire work. J.F.D.B. and B.G.Z. wrote the manuscript. All authors contributed to manuscript revision and editing.

## Competing interests

The authors declare no competing interests.
