## [Peer Review File · Communications Biology]

Reviewers' comments:

Reviewer #1 (Remarks to the Author):

The paper by Delgado-Blas et al. provides some very interesting results regarding E. coli population genomics in wastewater and river environments. Overall, the paper is well written and straightforward, the methods are robust, and the results provide important insights. The authors provide evidence of significant differences between E. coli populations retrieved from anthropogenic and natural water ecosystems. These findings are novel and of general interest to others in the community. Also, analyses are described in sufficient detail to guarantee reproducibility. However, the authors should take in consideration a few major points and some minor revisions.

Major revisions:

- The authors are assuming that each replicon corresponds to a single plasmid; however, it's common to see hybrid ('mosaic') plasmids, that can carry more than one replicon. Also, it would be interesting to look for other important plasmid features, such as the relaxase (MOB) family and other genes involved in conjugation, to understand if these elements can be mobilized or not;
- Even though the focus of this work was on the plasmid content, the authors should be aware that functions related to e.g. metal metabolism may be present in other mobile elements, such as genomic islands or ICEs (besides the metal resistance genes found on the InCHI2 plasmids). This manuscript already provides significant findings regarding the plasmid content and further analyses on this subject shouldn't be necessary, but I find it important that the authors discuss the possible contribution of these chromosomally integrated mobile elements to explain the higher genetic diversity found on river isolates;
- I couldn't find the information regarding the bioprojects and accession numbers provided throughout the manuscript. I understand the authors may want to release the projects to public only when the paper is published, but it's very important that they share this information with the reviewers;

Minor revisions:

- Line 108. Given the importance played by K. pneumoniae, don't the authors think that it's scarcity among WWTPs is weird? The authors could provide a possible explanation for this in the discussion section;
- Line 151/152. Any idea why these isolates were non-typeable by PFGE?
- Line 163. Similar MIC values in general, or only between isolates from the same ST? This is not clear here;
- Lines 210/211. What about genes belonging to STs shared by both origins? For example ST224 is found both in WWTPs and river samples;
- Line 275. Since the authors performed hybrid assembly for a set of isolates, it would be interesting to provide more detail on the genetic environment of the mdxA - is this located in a transposon, genomic island or ICE?
- Line 360. First time the htd gene appears; please provide more detail on its function;
- Line 390. This link is not working, even though I was able to browse through all of the other microreact projects;
- Line 420. The way this sentence is written, it gives the impression that all tested isolates were resistant to aminoglycosides. Even though the authors explain this before, it's important to rephrase

this sentence, highlighting the focus of this work on E. coli isolates positive for the 16S-RMTase gene;

- Line 430/431 and lines 462/463. In line with what was aforementioned in major revisions, the larger chromosomes verified in river isolates are also indicative of the presence of chromosomally integrated mobile elements, such as genomic islands, ICEs and prophages. Even though the focus of this work is on plasmids, the authors should acknowledge the possible presence of these mobile elements. The authors only briefly mention the presence of phage functions (line 265/267);
- Methods section. Taking a look at the tools' versions, it's possible to predict that some analyses were done quite a few years ago, or the pipeline wasn't properly updated. For example, the authors used spades v3.11, which was developed in 2017. The latest release is the 3.14. The same happens for prokka, unicycler and other tools.
- Line 626. Again, I cannot access this page;
- Line 643/645. Please clarify if default or optimized parameters were used here in unicycler;

Thank you for this manuscript, it was a pleasant reading.

Best wishes,
João Botelho

Reviewer #2 (Remarks to the Author):

In the submitted manuscript, Delgado-Blas et al. analyzed the population genomics and antimicrobial resistance dynamics of *Escherichia coli* in wastewater and river environments. The topic (AMR in aquatic environments) which covers is important, wet-lab and bioinformatic analyses are also appropriate. I have several concerns on the experimental design (e.g. the limited number of samples/isolates involved in this work) and data interpretation (i.e. argument about E. coli is the predominant species exhibiting pan-aminoglycoside resistance).

Major concerns:

1. My biggest concern would be the limited sample size. The authors were making a big argument about population genomics of E. coli from two eco-systems (i.e. wastewater and natural river), but only 8 (rivers)+5 (WWTPs)=13 samples were collected for analysis. How confident are you for your conclusion based on this limited number of samples? Or did you perform any power analysis before the sampling effort? 43 out of 70 E. coli isolates were selected for sequencing and downstream analysis, what's the rationale for making this selection?
2. Line 104-106 (and many other places): I'm not convinced this is a solid conclusion. All the isolates were recovered (artificially selected) on MacConkey media, and I would argue most recovered isolates would be E. coli. this is simply because that MAC agar is a selective media and has been commonly used to select for presumptive E. coli strains, this biased selection cannot support the idea E. coli is indeed the predominant species demonstrating pan-aminoglycoside resistance. Are E. coli, Enterobacter, and Citrobacter equally grow well on MacConkey agar?

Specific comments:

1. figure 1. I know all the isolates details were provided in supplementary tables, but if you could somehow shadow this information in the figure it would be very helpful for readers to catch the flow of your story (i.e. briefly state which/what isolates from which source were used for which analysis)
2. Line 68 (and other places): what's the point for capitalizing public health?

3. Line 70-72: the transition here is very weak. I still did not get the point why studying aminoglycosides resistance in aquatic environments is critical? I understand aquatic environments are generally a reservoir of AMR and of course aminoglycosides resistance is a clinical concern, but why aminoglycosides resistance in aquatic environments is a priority? Is it more prevalent in water or what?

Reviewer #3 (Remarks to the Author):

The manuscript presented by Delgado-Blas et al. is well written and comprises a very thorough study on antibiotic resistance in environmental *Escherichia Coli* strains. While this manuscript focuses only on a small part of the global antibiotic resistance problem (aminoglycosides in *E.coli*) this allows the authors to analyse this part in very high detail, including the global dissemination of such strains, combining a variety of wetlab and bioinformatic methods, which creates a lot of valuable knowledge for specialists in this field. From a more general point of view, it is interesting to see that the anthropogenic effect of causing less diversity and higher specialization on resistances in microorganisms can also be seen when zooming in on strains of a single species carrying the same antibiotic resistance, although this effect seems to be less clear when also considering the sequence type variety of both environments.

I have only a few additional questions and comments considering the length of the manuscript:

INTRODUCTION

1) Line 58-60: Something feels off in this sentence, please remove “[...]to unspoiled niches” or rephrase/describe in more detail if you consider this point absolutely necessary.

2) Line 68: as reference 2 is slightly outdated (2008), I would recommend an additional more recent reference for antibiotics-resistance dissemination between natural and anthropogenic environments.

For example, but not compulsory: Hernando-Amado, S., Coque, T.M., Baquero, F. et al. Defining and combating antibiotic resistance from One Health and Global Health perspectives. *Nat Microbiol* 4, 1432–1442 (2019). <https://doi.org/10.1038/s41564-019-0503-9>

3) Line 77: insert “and” or “which” or similar between “[...] group have[...]”

RESULTS

4) Fig.1: please add the number of isolates/sample origin in Fig.1 (for example as number in the middle of the circle diagrams)

5) In the methods line 511/512 you describe that you did PCR tests for a variety of 16S-RMTase genes, but in your results line 114ff. you report only the detection of *armA* and *rmtB*. Does that mean all of your isolates were negative for *rmtA*, *rmtC*, *rmtD*, *rmtE*, *rmtF*, *rmtG*, *rmtH* and *npmA* according to your PCR tests or did you only check for two genes? Please clarify.

6) Line 264: “[...]functions linked to regulation[...]” I guess you mean “linked to” here

7) Figure 5: The references are stated a bit in a confusing manner for me here:. Why does the IncFII comparison have twice the isolate BB1418 but only one of it is marked as reference? Maybe it would be clearer when you define external references in the “source” column of the small tables. Please

also add a citation for the reference publications in the legend of fig.5.

8) Line 354: “[...]these IncHI2A plasmid[...]” ◊ “[...]these IncHI2A plasmids[...]”

9) Line 379/380: The supplementary .KML file is provided only as .XML file to me but after renaming the file ending it worked without problems. Also, your reference 26 links to the Spanish version of google Earth, you should replace “/es/” with “/en/” in your reference.

Additional Comment: I think it is a very nice idea to add all these supplementary interactive trees and maps to a publication by the way, I hope these websites are available for a long time.

DISCUSSION

10) Line 411/412: As *E. coli* is an indicator of fecal contamination, the presence of *E. coli* in rivers (especially of antibiotic resistant strains) might also be an indicator for upstream pollution by agriculture (i.e. influx of liquid manure), what is your stance regarding this possibility? Maybe you can shortly discuss this in the manuscript.

METHODS

11) The methods are well written and in sufficient detail, citing all used bioinformatics tools and describing used thresholds and other parameters as well as statistical tests clearly.

I cannot check given ENA accession numbers or the code on github, as they are not yet publicly available, but as several accession numbers are given in the supplementary, it looks like all the data is already properly submitted. I trust these will be publicly available before publication

12) I am confused regarding the amount of samples you took:

Line 479/480: “A total of 8 samples were obtained from two rivers[...]”

Line 484/485: “From each river, samples were taken from surface water (3 samples) and sediments (1 sample) at two locations and on different dates.”

Does this mean you took 8 samples from each river, meaning actually 16 in total, as you sampled 3+1 samples for each of two different locations per river? Or did you sample 3 water samples from one location and 1 sediment sample from another location for each river?

Ideally please add a supplementary table containing all your original samples which includes also the (approximate) coordinates of the sampling sites to prevent any confusion in this regard. Coordinates additionally help to put the sampling sites in a better perspective (e.g. do the samples originate more from the rural or urban part of the rivers? Were the Llobregat samples taken rather physically close or distant to the WWTPs, and before or after El Cardener joins El Llobregat? Were the river samples taken physically close to each other?) Alternatively to the coordinates you could also mark the sampling spots in Figure 1 more clearly.

13) Line 512: reference 34 provides the primers but not the PCR conditions for 16S-RMTase gene screening, please provide all necessary information to replicate your experiments.

14) Line 541 ff. “Whole-genome sequencing by Illumina and data processing[...]”

How did you handle removal of Illumina adapters? Was it already done by the sequencing company?

15) Line 765 ff: “Author contributions[...]”

One author (N.M.) is not explicitly mentioned, is this intended?

Response to referees' comments

Please find enclosed our response to the referees' comments. I would like to underline that they were very helpful, resulting in a, in my eyes, clearer manuscript now.

MANUSCRIPT: "Population genomics and antimicrobial resistance dynamics of *Escherichia coli* in wastewater and river environments"
COMMSBIO-20-2226-T

Reviewer #1 (Remarks to the Author):

The paper by Delgado-Blas et al. provides some very interesting results regarding *E. coli* population genomics in wastewater and river environments. Overall, the paper is well written and straightforward, the methods are robust, and the results provide important insights. The authors provide evidence of significant differences between *E. coli* populations retrieved from anthropogenic and natural water ecosystems. These findings are novel and of general interest to others in the community. Also, analyses are described in sufficient detail to guarantee reproducibility. However, the authors should take in consideration a few major points and some minor revisions.

Major revisions:

1. The authors are assuming that each replicon corresponds to a single plasmid; however, it's common to see hybrid ('mosaic') plasmids, that can carry more than one replicon. Also, it would be interesting to look for other important plasmid features, such as the relaxase (MOB) family and other genes involved in conjugation, to understand if these elements can be mobilized or not;

The presence of hybrid ('mosaic') plasmids was analyzed and determined. The results and their possible implications derived from them have been included in the manuscript (**Line 345**).

The key genetic elements involved in plasmid conjugation, including relaxase (MOB), T4CP and VirB4 genes, together with other multiple accessory genes related with conjugative systems, were identified and classified in order to analyze the potential capacity of conjugation of the 16S-RMTase gene-carrying plasmids. The results have been included in the manuscript (**Line 461**), as well as the methods performed to obtain them (**Line 809**).

2. Even though the focus of this work was on the plasmid content, the authors should be aware that functions related to e.g. metal metabolism may be present in other mobile elements, such as genomic islands or ICEs (besides the metal resistance genes found on the IncHI2 plasmids). This manuscript already provides significant findings regarding the plasmid content and further analyses on this subject shouldn't be necessary, but I find it important that the authors discuss the possible contribution of

these chromosomally integrated mobile elements to explain the higher genetic diversity found on river isolates;

The metal metabolism functions specifically identified in river *E. coli* isolates were encoded by gene clusters directly related with the metal resistance genes carried by the IncHI2A plasmids present in these isolates. None of these functions was associated with other mobile genetic elements. This point has been clarified in the manuscript (**Line 448**).

We have performed additional analyses to address the cause of the higher genetic diversity found on the chromosome of river isolates, determining the responsible genetic elements, including chromosomally integrated mobile elements. These results and their implications have been included in the manuscript (**Lines 330 and 556**), as well as the methods performed to obtain them (**Line 795**).

3. I couldn't find the information regarding the bioprojects and accession numbers provided throughout the manuscript. I understand the authors may want to release the projects to public only when the paper is published, but it's very important that they share this information with the reviewers;

All sequencing data submitted to the European Nucleotide Archive (ENA) is ready to be released and publicly available under the umbrella project PRJEB34801 after acceptance for publication.

Minor revisions:

1- Line 108. Given the importance played by *K. pneumoniae*, don't the authors think that it's scarcity among WWTPs is weird? The authors could provide a possible explanation for this in the discussion section;

The prevalence of aminoglycoside-resistant *K. pneumoniae* in WWTPs has been addressed and a possible explanation has been included in the discussion of the manuscript (**Line 540**).

2- Line 151/152. Any idea why these isolates were non-typeable by PFGE?

All non-typable *E. coli* by PFGE belonged to the same sequence type, ST224. The procedure was done together with other isolates belonging to other STs and controls that were properly typed, discarding the possibility of technical errors. The “non-typeable” phenomenon in this case could be due to multiple biological reasons, maybe a specific genomic conformation of this ST isolates that led to an irregular degradation of the genetic material by the restriction enzymes, resulting in the smear pattern instead of the band pattern needed for accurate typing.

3- Line 163. Similar MIC values in general, or only between isolates from the same ST? This is not clear here;

Similar MIC values in general, including all river isolates belonging to different STs. It has been clarified in the manuscript (Line 189).

4- Lines 210/211. What about genes belonging to STs shared by both origins? For example ST224 is found both in WWTPs and river samples;

The Jaccard index and PERMANOVA values have been specifically calculated for ST224 *E. coli* from both origins. The results confirmed the statistical differences between the pangenome composition of isolates from wastewater and river environments. These results have been included in the manuscript (Line 249).

5- Line 275. Since the authors performed hybrid assembly for a set of isolates, it would be interesting to provide more detail on the genetic environment of the *mdfA* - is this located in a transposon, genomic island or ICE?

The gene *mdfA* is commonly found in the chromosome of *E. coli*, as it is pointed out in the manuscript, and it has been even considered an intrinsic-like resistance gene of this bacterial species, which highlights its close and ancient association. Besides, we performed a detailed characterization of the genetic environment of *mdfA* in the chromosome of *E. coli* isolates with available hybrid assembly, confirming that the gene was flanked by the same chromosome-related genes in all the cases, and no mobile genetic element was identified close to *mdfA*. These findings were included in the manuscript (Line 339).

6- Line 360. First time the *htd* gene appears; please provide more detail on its function;

A more detailed description of *htd* and *trh* genes has been including in the manuscript, together with their biological interactions and their role in the conjugation process of IncHI plasmids (Line 452).

7- Line 390. This link is not working, even though I was able to browse through all of the other microreact projects;

The link for this Microreact project has been checked and it is working correctly and publicly available (<https://microreact.org/project/a9-1Zlze0>). Line 503

8- Line 420. The way this sentence is written, it gives the impression that all tested isolates were resistant to aminoglycosides. Even though the authors explain this before, it's important to rephrase this sentence, highlighting the focus of this work on *E. coli* isolates positive for the 16S-RMTase gene;

The sentence has been rephrased, clarifying this point in the manuscript (Line 537).

9- Line 430/431 and lines 462/463. In line with what was aforementioned in major revisions, the larger chromosomes verified in river isolates are also indicative of the presence of chromosomally integrated mobile elements, such as genomic islands, ICEs and prophages. Even though the focus of this work is on plasmids, the authors should

acknowledge the possible presence of these mobile elements. The authors only briefly mention the presence of phage functions (line 265/267);

This point has been previously addressed in the major comments, following the reviewer's suggestion. Additional analyses were carried out to identify the genetic elements involved in chromosome variations in river *E. coli* isolates. We have determined that most of the chromosomally integrated mobile elements responsible for these variations included insertion sequences, transposons and phage-related elements. These results have been specified in the manuscript (**Lines 330, 337 and 556**), as well as the methods performed to obtain them (**Line 795**).

10- Methods section. Taking a look at the tools' versions, it's possible to predict that some analyses were done quite a few years ago, or the pipeline wasn't properly updated. For example, the authors used spades v3.11, which was developed in 2017. The latest release is the 3.14. The same happens for prokka, unicycler and other tools.

The tools' versions specified in the manuscript are those used to perform the different analyses. Most of these tools are server-based and integrated as part of pipelines that have specific requirements for downstream analyses. Thus, using the specific versions for which these pipelines were developed is fundamental to allow the accuracy and reproducibility of the results.

11- Line 626. Again, I cannot access this page;

The GitHub repository with the R codes designed for genomic diversity in bacterial populations is open and publicly available (https://github.com/JoseFranciscoDelgadoBlas/bacterial_population_diversity). **Line 764**

12- Line 643/645. Please clarify if default or optimized parameters were used here in unicycler;

All hybrid assemblies performed with Unicycler were carried out by applying default parameters. This point was specified in the manuscript (**Line 784**).

Thank you for this manuscript, it was a pleasant reading.

**Best wishes,
João Botelho**

Reviewer #2 (Remarks to the Author):

In the submitted manuscript, Delgado-Blas et al. analyzed the population genomics and antimicrobial resistance dynamics of Escherichia coli in wastewater and river environments.

The topic (AMR in aquatic environments) which covers is important, wet-lab and bioinformatic analyses are also appropriate. I have several concerns on the experimental design (e.g. the limited number of samples/isolates involved in this work)

and data interpretation (i.e. argument about *E. coli* is the predominant species exhibiting pan- aminoglycoside resistance).

Major concerns:

1. My biggest concern would be the limited sample size. The authors were making a big argument about population genomics of *E. coli* from two eco-systems (i.e. wastewater and natural river), but only 8 (rivers)+5 (WWTPs)=13 samples were collected for analysis. How confident are you for your conclusion based on this limited number of samples? Or did you perform any power analysis before the sampling effort? 43 out of 70 *E. coli* isolates were selected for sequencing and downstream analysis, what's the rationale for making this selection?

The water samples analyzed in this study were collected from different environmental locations, covering a large area of around 50km². These sampling locations and the distribution of the samples were previously and specifically selected in order to obtain an accurate representation of the *Enterobacteriaceae* populations from natural river and wastewater treatment plant environments in the area, including a total of 3 different points in the course of the two main rivers and a total of 5 wastewater tanks of the two main WWTPs (specified in the Methods of the manuscript, "Sampling and bacterial isolation" section **Line 607**). Furthermore, the samples were taken on different dates to counterbalance the possible sampling time-related bias. The distribution of the bacterial species identified in the different sampling locations (Figure 1 in the manuscript and Supplementary Table 1 of the Supplementary Data), among other markers, confirmed that the samples covered a suitable representation of the *Enterobacteriaceae* populations from both environments. This distribution in the two WWTPs was highly similar, even when they collect wastewater from two areas of the sampled region, revealing a common *E. coli* composition across the whole population of the region. The distribution of bacterial species in river samples showed a consistent pattern throughout the river courses, showing the variability of river environments across space and time. No specific power statistical analysis was performed prior to the sampling process, but the predefined sampling conditions together with the bacterial markers analyzed afterwards demonstrated the suitability of the collected water samples for subsequent analyses.

The selection of the 43 out of the total 70 *E. coli* isolates recovered from water samples for whole genome sequencing and posterior analyses was based on the origin of the isolates (sampling point and location), its clonal relatedness by PFGE and their resistance profiles assessed by antibiograms to different antimicrobial compounds and families, in order to avoid clonal repetitiveness. This selection criteria are described in the methods (**Line 671**), as well as in the results of the manuscript (**Line 163**).

2. Line 104-106 (and many other places): I'm not convinced this is a solid conclusion. All the isolates were recovered (artificially selected) on MacConkey media, and I would argue most recovered isolates would be *E. coli*. this is simply because that MAC agar is a selective media and has been commonly used to select for presumptive *E. coli* strains, this biased selection cannot support the idea *E. coli* is indeed the predominant species demonstrating pan-aminoglycoside resistance. Are *E. coli*, *Enterobacter*, and *Citrobacter* equally grow well on MacConkey agar?

MacConkey medium is a selective culture medium for Gram-negative enteric bacilli, mainly for those belonging to the *Enterobacteriaceae* family, including species of the genus *Escherichia*, *Enterobacter* and *Citrobacter* among others, but also for other Gram-negative bacilli from other families, such as the *Aeromonadaceae* or *Pseudomonadaceae*. All these bacterial species are equally able to grow in this medium. Since the MacConkey medium used in the bacterial isolation procedure for this study was supplemented with aminoglycosides at high concentrations (specified in the Methods of the manuscript, “Sampling and bacterial isolation” section **Line 616**), strictly talking, we selected all Gram-negative enteric bacilli, mainly belonging to *Enterobacteriaceae* species, exhibiting pan-aminoglycoside resistance. In fact, we have applied this bacterial isolation method for other studies focused on antimicrobial resistance in wastewater bacterial populations where the predominant bacterial species belonged to other bacterial genus and family. Taking into account the reviewers’ comment, we have specified in all concerning sentences of the manuscript that *E. coli* was the “predominant pan-aminoglycoside resistant *Enterobacteriaceae* species”, instead of the “predominant pan-aminoglycoside resistant species” (e.g. **Lines 115 and 526**).

Specific comments:

1. figure 1. I know all the isolates details were provided in supplementary tables, but if you could somehow shadow this information in the figure it would be very helpful for readers to catch the flow of your story (i.e. briefly state which/what isolates from which source were used for which analysis)

Figure 1 has been modified, indicating the total number of pan-aminoglycoside resistant isolates and the number of the most prevalent bacterial species obtained in each sampling location. Likewise, the number of pan-aminoglycoside resistant *E. coli* from each sampling location selected for later Illumina and Nanopore sequencing and analysis has been also specified in the figure.

Figure 1. Geographical distribution of sampled points in the region of Barcelona (Spain). The location of Barcelona city is indicated by 🏙️. The Llobregat river is presented in light blue and the Cardener river in dark blue. The three river sampled locations are indicated by 📍. The two sampled WWTPs are indicated by 🏭. The collection area of the El Prat WWTP is highlighted in dark green, whereas the area of the Gavà WWTP is highlighted in pale green, collecting the wastewater of 2,000,000 and 370,000 inhabitants, respectively. Total number of pan-aminoglycoside resistant bacteria collected from each sampling location is indicated in the center of the sunburn diagrams. Inner rings represent the proportion of different pan-aminoglycoside resistant bacterial species identified in each sampling location, indicating the total number of *E. coli*, *K. pneumoniae* and other species, and the number of *E. coli* isolates selected for later Illumina ^(I) and Nanopore sequencing ^(N). Outer rings show the 16S-RMTase gene harbored by these bacteria.

2. Line 68 (and other places): what's the point for capitalizing public health?

“Public Health” was capitalized to emphasize the relevance of the concept. However, it has been modified and rewritten with lower case letters (“public health”) across the manuscript.

3. Line 70-72: the transition here is very weak. I still did not get the point why studying aminoglycosides resistance in aquatic environments is critical? I understand aquatic environments are generally a reservoir of AMR and of course aminoglycosides resistance is a clinical concern, but why aminoglycosides resistance in aquatic environments is a priority? Is it more prevalent in water or what?

The emergence and dissemination of bacteria resistant to other last-resort antibiotics, such as carbapenems and polymyxins, have positioned the aminoglycosides as one of the main antimicrobial groups in the fight against clinical multi-drug resistant bacteria. Therefore, the resistance to aminoglycosides, and its association with other resistance mechanisms, has become a critical public health concern. As the reviewer points out and it is explained in the manuscript, aquatic environments are important reservoirs of AMR, including aminoglycoside resistance and 16S rRNA methyltransferases. However, the situation of these resistance determinants, their carrying mobile genetic elements and their producing bacteria is largely unknown in both anthropogenic and natural environments, but particularly in the latter ones. This is mainly due to the lack of works addressing this subject, contrasting with other antimicrobial groups. For all these reasons, the study of the epidemiological scenario of aminoglycoside resistance determinants, especially the 16S rRNA methyltransferases, in water environments is a priority to determine their potential impact in public health and critical to establish measures aimed at the preservation of this important antimicrobial group effectiveness. The Introduction of the manuscript has been modified to include a clarifying explanation in order to determine the relevance of the study of these antibiotic resistance mechanisms in aquatic environments (**Line 95**).

Reviewer #3 (Remarks to the Author):

The manuscript presented by Delgado-Blas et al. is well written and comprises a very thorough study on antibiotic resistance in environmental Escherichia Coli strains. While this manuscript focuses only on a small part of the global antibiotic resistance problem (aminoglycosides in E.coli) this allows the authors to analyse this part in very high detail, including the global dissemination of such strains, combining a variety of wetlab and bioinformatic methods, which creates a lot of valuable knowledge for specialists in this field. From a more general point of view, it is interesting to see that the anthropogenic effect of causing less diversity and higher specialization on resistances in microorganisms can also be seen when zooming in on strains of a single species carrying the same antibiotic resistance, although this effect seems to be less clear when also considering the sequence type variety of both environments.

I have only a few additional questions and comments considering the length of the manuscript:

INTRODUCTION

1) Line 58-60: Something feels off in this sentence, please remove “[...]to unspoiled niches” or rephrase/describe in more detail if you consider this point absolutely necessary.

Part of the sentence has been removed in the manuscript (**Line 63**).

2) Line 68: as reference 2 is slightly outdated (2008), I would recommend an additional more recent reference for antibiotics-resistance dissemination between natural and anthropogenic environments.

For example, but not compulsory: **Hernando-Amado, S., Coque, T.M., Baquero, F. et al. Defining and combating antibiotic resistance from One Health and Global Health perspectives. Nat Microbiol 4, 1432–1442 (2019). <https://doi.org/10.1038/s41564-019-0503-9>**

The suggested reference has been included in the manuscript (**Line 72**).

3) Line 77: insert “and” or “which” or similar between “[...] group have[...]”

The sentence has been changed in the manuscript according to the reviewer’s suggestion.

RESULTS

4) Fig.1: please add the number of isolates/sample origin in Fig.1 (for example as number in the middle of the circle diagrams)

The total number of pan-aminoglycoside resistant isolates and the number of the most prevalent bacterial species obtained in each sampling location have been added in Figure 1. See **Reviewer #2, Specific comment 1**.

5) In the methods line 511/512 you describe that you did PCR tests for a variety of 16S-RMTase genes, but in your results line 114ff. you report only the detection of armA and rmtB. Does that mean all of your isolates were negative for rmtA, rmtC,

rmtD, rmtE, rmtF, rmtG, rmtH and npmA according to your PCR tests or did you only check for two genes? Please clarify.

The isolates were tested for all known 16S-RMTase genes by PCR and they were negative for all genes other than *armA* and *rmtB*. This point has been clarified in the Results of the manuscript (Line 146).

6) Line 264: “[...]functions linked to regulation[...]” I guess you mean “linked to” here

The reviewer is right. The sentence has been corrected in the manuscript (Line 311).

7) Figure 5: The references are stated a bit in a confusing manner for me here:. Why does the IncFII comparison have twice the isolate BB1418 but only one of it is marked as reference? Maybe it would be clearer when you define external references in the “source” column of the small tables. Please also add a citation for the reference publications in the legend of fig.5.

The IncFII comparison have twice the isolate BB1418 because this isolate harbored two different plasmids belonging to IncFII plasmid type. Only one of them is marked as reference because one of them (the one belonging to the specific pHN7A8 IncFII type) was applied as the common reference for the comparison of all plasmids belonging to IncFII type. All plasmids used for structural plasmid comparisons were obtained in the present work. All this information is suitably indicated in the legend of Figure 5 (Line 388) and specified in the Methods of the manuscript (“16S-RMTase gene-carrying plasmid analysis” section, Line 803).

8) Line 354: “[...]these IncHI2A plasmid[...]” ◊ “[...]these IncHI2A plasmids[...]”

The reviewer is right. The sentence has been corrected in the manuscript (Line 446).

9) Line 379/380: The supplementary .KML file is provided only as .XML file to me but after renaming the file ending it worked without problems. Also, your reference 26 links to the Spanish version of google Earth, you should replace “/es/” with “/en/” in your reference.

The supplementary KML file (Supplementary_file_ST1196_dissemination_interactive_map.kml) was uploaded with the .kml extension (Line 489). The extension change should be due to a file transfer modification. We appreciate the reviewer’s advice.

The reference link for Google Earth has been modified to the English version (Google Earth Pro. *Google Earth* <https://www.google.com/intl/en/earth/>). Line 1021

Additional Comment: I think it is a very nice idea to add all these supplementary interactive trees and maps to a publication by the way, I hope these websites are available for a long time.

We appreciate the reviewer's comment. These all supplementary interactive trees and maps should be permanently available, since the host websites support the maintenance of all these data and visualizations, even after future version updates.

DISCUSSION

10) Line 411/412: As *E. coli* is an indicator of fecal contamination, the presence of *E. coli* in rivers (especially of antibiotic resistant strains) might also be an indicator for upstream pollution by agriculture (i.e. influx of liquid manure), what is your stance regarding this possibility? Maybe you can shortly discuss this in the manuscript.

This point was previously addressed in the manuscript, in the section “*E. coli* STs have a similar genomic complexity in wastewater and river water, but the diversity of plasmids and resistance genes is higher in wastewater STs” of the Results and in the Discussion. The animal or even human related origin of the *E. coli* population identified in river environments cannot be totally rejected. However, multiple indicators support the environmental nature of these *E. coli* isolates. Firstly, previous studies determined the minor impact of anthropogenic activities in the higher course of sampled rivers, as mentioned in the Methods of the manuscript (“Sampling and bacterial isolation” section). Most importantly, all phenotypic, genomic and diversity analyses performed throughout the study have demonstrated significant differences between both wastewater and river *E. coli* populations, even for the sequence type present in both environments. Taking into account that the two sampled WWTPs collect the wastewater from the human and animal populations inhabiting the region, if *E. coli* river isolates were originated in these populations, they should be similar, if not identical, to *E. coli* isolates identified in wastewater. Furthermore, some of the *E. coli* isolates analyzed in river waters were identified in time points separated by more than 2 months, indicating the colonization and establishment of these bacteria in the natural environment. Even if the primordial origin of river isolates was animal or human related, the persistence of these bacteria in natural environments and the significant differences with residual bacteria in the strict sense are clear evidences of the naturalization of river *E. coli* and, therefore, of their environmental nature. All these aspects supporting the environmental origin of *E. coli* in river water have been briefly summarized and included in the Discussion of the manuscript (**Line 530**).

METHODS

11) The methods are well written and in sufficient detail, citing all used bioinformatics tools and describing used thresholds and other parameters as well as statistical tests clearly.

I cannot check given ENA accession numbers or the code on github, as they are not yet publicly available, but as several accession numbers are given in the supplementary, it looks like all the data is already properly submitted. I trust these will be publicly available before publication

As previously mentioned, all sequencing data submitted to the European Nucleotide Archive (ENA) is ready to be released and publicly available under the umbrella project PRJEB34801 after acceptance for publication.

12) I am confused regarding the amount of samples you took:

Line 479/480: “A total of 8 samples were obtained from two rivers[...]”

Line 484/485: “From each river, samples were taken from surface water (3 samples) and sediments (1 sample) at two locations and on different dates.”

Does this mean you took 8 samples from each river, meaning actually 16 in total, as you sampled 3+1 samples for each of two different locations per river? Or did you sample 3 water samples from one location and 1 sediment sample from another location for each river?

The total number of collected river samples was 8, comprising the two rivers. From the Llobregat river, the main water source of the region, 6 samples were taken at two different locations: in location A (located in the middle course of the river and, therefore, closer to the areas served by the sampled WWTPs), 3 surface water samples were collected on different dates and 1 sample was obtained from the river sediments; in location B (located in the high-middle course of the river), 1 surface water sample and 1 sediments sample were collected. Regarding the Cardener river, a tributary of Llobregat river that flows into it in its high-middle course, 2 samples were taken from one location, 1 surface water sample and 1 sediments sample (similar to location B in the Llobregat river). This strategy and distribution for river sampling was designed in order to obtain an overall picture of the bacterial populations from river environments of the region, but focusing particularly in the main water body, the Llobregat river, and more specifically in the stretch of the river located in the area from which the sampled WWTPs collect the wastewater. The point of this strategy was the collection of a suitable representation of bacteria from both residual and natural water environments, which share a common geographical area, allowing its phenotypic, genomic and diversity comparison. The sampling strategy in river environments has been extended and clarified in the manuscript, in the “Sampling and bacterial isolation” section of the Methods (**Line 613**).

Ideally please add a supplementary table containing all your original samples which includes also the (approximate) coordinates of the sampling sites to prevent any confusion in this regard. Coordinates additionally help to put the sampling sites in a better perspective (e.g. do the samples originate more from the rural or urban part of the rivers? Were the Llobregat samples taken rather physically close or distant to the WWTPs, and before or after El Cardener joins El Llobregat? Were the river samples taken physically close to each other?) Alternatively to the coordinates you could also mark the sampling spots in Figure 1 more clearly.

Geographical coordinates of all collected samples from river and wastewater environments are indicated in Supplementary Table 2a of the Supplementary Data, specified as metadata of *E. coli* isolates obtained from the samples. Figure 1 has been modified in order to show more clearly the specific locations of the sampling points in the schematic map of the region (See **Reviewer #2, Specific comment 1**). In addition, an interactive map with the geographical distribution of the sampling points from where *E. coli* isolates belonging to the most prevalent STs were identified in this study can be visualized in the Microreact link: <https://microreact.org/project/a9-1Zlze0> (**Line 503**).

The maps, interactive visualizations and supplementary material provided with the manuscript allow to put in perspective the sampling points, as the reviewer mentioned. All river samples were collected from sections of the rivers considered rural environments, in

the high and middle course, avoiding the lower course, which is located in an urban area and close to the WWTPs. Thus, Llobregat samples were taken physically distant of the WWTPs: approximately 35km between the WWTPs and Llobregat location A, and 50km between the WWTPs and Llobregat location B. Both Llobregat sampling locations were located after the Cardener river flows into it to assess the possible influence of bacteria from Cardener river in the bacterial population of Llobregat river at different sections. River samples were taken physically distant between them to obtain an overall picture of bacterial populations from river environments of the region: approximately 35km between the Cardener sampling point and Llobregat sampling location B and 15km between the latter one and Llobregat sampling location A.

13) Line 512: reference 34 provides the primers but not the PCR conditions for 16S-RMTase gene screening, please provide all necessary information to replicate your experiments.

The referenced work provides all necessary information to replicate the 16S-RMTase gene screening by PCR, including primer sequences, melting temperature and product size for each of the screened genes. Furthermore, the work also indicates the different studies from which the PCR protocols were obtained (**Line 1042**).

**14) Line 541 ff. “Whole-genome sequencing by Illumina and data processing[...]”
How did you handle removal of Illumina adapters? Was it already done by the sequencing company?**

The removal of Illumina adapters was performed with the tool Trimmomatic version 0.33, together with the trimming of end nucleotides with low sequencing quality. This point has been indicated in the manuscript, in the “Whole-genome sequencing by Illumina and data processing” of the Methods (**Line 682**).

**15) Line 765 ff: “Author contributions[...]”
One author (N.M.) is not explicitly mentioned, is this intended?**

The reviewer is right. The “Author contributions” section has been modified and N.M. (Natalia Montero) has been appropriately included (**Lines 922 and 924**).

Yours sincerely,

Bruno González Zorn

Head of the Antimicrobial Resistance Unit (ARU)
Faculty of Veterinary Medicine and VISAVET
Complutense University of Madrid

REVIEWERS' COMMENTS:

Reviewer #1 (Remarks to the Author):

The authors have properly addressed all comments from both reviewers.

Reviewer #2 (Remarks to the Author):

Thank you for addressing my concerns, no further comments.

Reviewer #3 (Remarks to the Author):

All my concerns have been addressed very well and I consider the manuscript in its current state suitable for publication.

Minor comment which should be easily mended: the interactive map at <https://microreact.org/project/a9-1Zlze0> shows only one Llobregat sampling point and the two WWTPs, there should be two more sampling points (Cardener & Llobregat) according to suppl. table 2A.

Response to referees' comments

Please find enclosed our response to the referees' comments. We thank the reviewers for their help and for their comments, which have contributed to enhance this work.

MANUSCRIPT: "Population genomics and antimicrobial resistance dynamics of *Escherichia coli* in wastewater and river environments"
COMMSBIO-20-2226-B

Reviewer #1 (Remarks to the Author):

The authors have properly addressed all comments from both reviewers.

Reviewer #2 (Remarks to the Author):

Thank you for addressing my concerns, no further comments.

Reviewer #3 (Remarks to the Author):

All my concerns have been addressed very well and I consider the manuscript in its current state suitable for publication.

Minor comment which should be easily mended: the interactive map at <https://microreact.org/project/a9-1Zlze0> shows only one Llobregat sampling point and the two WWTPs, there should be two more sampling points (Cardener & Llobregat) according to suppl. table 2A.

Interactive Microreact map indicated by the reviewer shows the worldwide distribution of isolates belonging to **the three most-prevalent 16S-RMTase gene-carrying *E. coli* STs** identified in waters of Barcelona: ST1196, ST224 and ST607. No isolates belonging to these STs were detected in the sampling points mentioned by the reviewer (Llobregat B and Cardener river) and, therefore, these sampling points do not appear in the map. Supplementary Data 2a file includes the metadata and sequencing data of all 16S-RMTase gene-carrying *E. coli* isolates obtained from water samples in the region of Barcelona and sequenced by WGS Illumina technology, as indicated in the Supplementary Data caption and the manuscript. For this reason, this data file includes *E. coli* isolates from the sampling points mentioned by the reviewer, which belong to other STs.

The information of all *E. coli* isolates included in this and all interactive maps of the study is specified in the description of the projects at Microreact website, as well as in the Results and Methods sections of the manuscript and the related Supplementary Data files, which indicate the metadata and sequencing data of *E. coli* isolates included in the different analyses of the study.

Yours sincerely,

BRUNO GONZALEZ ZORN

Bruno González Zorn

Head of the Antimicrobial Resistance Unit (ARU)
Faculty of Veterinary Medicine and VISAVET
Complutense University of Madrid